# Generalized leaky integrate-and-fire models classify multiple neuron types

Corinne Teeter [1], Ramakrishnan Iyer [1], Vilas Menon[1,2], Nathan Gouwens[1], David Feng [1], Jim Berg[1], Aaron Szafer[1], Nicholas Cain [1], Hongkui Zeng[1], Michael Hawrylycz[1], Christof Koch [1] & Stefan Mihalas [1]

There is a high diversity of neuronal types in the mammalian neocortex. To facilitate construction of system models with multiple cell types, we generate a database of point models associated with the Allen Cell Types Database. We construct a set of generalized leaky integrate-and-fire (GLIF) models of increasing complexity to reproduce the spiking behaviors of 645 recorded neurons from 16 transgenic lines. The more complex models have an increased capacity to predict spiking behavior of hold-out stimuli. We use unsupervised methods to classify cell types, and find that high level GLIF model parameters are able to differentiate transgenic lines comparable to electrophysiological features. The more complex model parameters also have an increased ability to differentiate between transgenic lines. Thus, creating simple models is an effective dimensionality reduction technique that enables the differentiation of cell types from electrophysiological responses without the need for a priori-defined features. This database will provide a set of simplified models of multiple cell types for the community to use in network models.

[1] Allen Institute for Brain Science, 615 Westlake Ave N, Seattle, WA 98109, USA. [2] Howard Hughes Medical Institute, Janelia Research Campus, 19700 Helix Dr, Ashburn, VA 20147, USA. Correspondence and requests for materials should be addressed to C.T. (email: corinnet@alleninstitute.org) or to S.M. (email: stefanm@alleninstitute.org)

The problem of understanding the complexity of the brain has been central to neuroscience. The classification of neurons into cell types is a conceptual simplification intended to reduce this complexity. To this end, a large-scale effort at the Allen Institute for Brain Science has focused on characterizing the diversity of cell types in the primary visual cortex of the adult mouse using electrophysiology, morphological reconstructions, connectivity, and modeling in one standardized effort. This has resulted in the Allen Cell Types Database[1] that is publicly and freely available at http://celltypes.brain-map.org. It includes both morphological and electrophysiological data of genetically identified neurons mapped to the common coordinate framework[2]. Morphologically and biophysically detailed as well as simple generalized leaky integrate-and-fire point neuron models have been generated[1] to reproduce cellular data produced under highly standardized conditions.

Creating simplified models is a way to reduce the complexity of the brain to its most fundamental mechanisms. In addition to the benefits of clarifying mechanisms for single-neuron behavior, single-neuron models can be used in larger network models that attempt to explain network computation. Thus, many models of a wide range of complexity have been developed to describe and recreate various aspects of neuronal behavior[3]. For an in-depth characterization of the diversity of neuron models, see the review[4], and for their capacity to reproduce spike times see ref. [5].

At the high end of the complexity spectrum, are the morphologically and biophysically realistic Hodgkin–Huxley-like models[6–8]. Their strength lies in their capacity to map between multiple observables: morphology, electrophysiology, intracellular calcium concentration, and levels of expression and patterns of distributions of ionic currents. Although adding complexity to a model may increase the ability of that model to recreate certain behavior, finding the right parameters for complex models becomes a challenge[9]. Furthermore, the computational power needed to simulate sophisticated neural models can be quite large[10]. Therefore, ideally one would use a computationally minimal model adequate to recreate and to understand the desired behavior[11]. One simplification that significantly reduces model complexity is to represent the entire dendritic tree, soma, and axonal hillock by a single compartment, while maintaining the dynamics of the individual conductances[3]. This approximation is especially warranted when characterizing neurons via somatic current injection and voltage recording as is done in the Allen Cell Types Database.

Here we report on the point neuron modeling portion of the Allen Cell Types Database[1]. In this study, we aimed to identify simple models that could both effectively reduce the biological space to a set of useful parameters for cell type classification and recreate spiking behavior for a diverse set of neurons for use in network models. In the adult cortex, the majority of communication between neurons is via chemical synapses from axons onto dendritic or somatic membrane (with a fraction of inhibitory neurons coupled by gap junctions as notable exceptions). The response of these non-NMDA synapses is generally dependent only on the action potentials generated by the presynaptic cell. Thus, we focus on reproducing the temporal properties of spike trains using computationally compact point neuron models. This spike-train focus allows us to generate models which are much simpler than biophysically detailed models but still capture a substantial amount of their complexity.

An extensive amount of work has gone into constructing simple models that can accurately reproduce the spiking behavior of neurons[5]. Yet many large network modeling papers use traditional leaky integrate and fire models (for example, ref. [12]). We wanted to understand how much may be gained (or lost) in adding complexity to such models. Thus, we characterized how adding phenomenological complexity to a model influences its ability to reproduce neuronal spike times and classify cells.

We used a family of models we refer to as generalized leaky integrate and fire (GLIF) models. This family of models starts with the classic leaky integrate and fire model[13] and then incorporates additional phenomenological mechanisms similar to other studies. These mechanisms are fit directly from the electrophysiological data and loosely represent slow activation/inactivation of voltage-gated ion channels and long-term effects of ion channel currents. Although there are many simplified and computationally compact models[14] available in the literature, the GLIF framework has a combination of advantages ideal for this study. First, GLIFs have been shown to be able to recreate a wide variety of biologically realistic behavior[15]. Second, the dynamical equations of the GLIF model are linear, and various versions have been previously fit to biological data[15–20]. In addition, phenomenological mechanisms loosely representing real biological mechanisms can be added to the LIF model that can easily be interpreted and potentially could be mapped to biological mechanisms. Other simplified non-linear models such as those of Izhikevich[21,22] can be difficult to fit to data[23] and are less phenomenologically interpretable. Furthermore, Mensi et al.[19] have shown the potential of GLIF models to classify cell types by identifying three cortical neuronal types using a linear classifier.

Here we expand on previous work, and apply GLIF-fitting tools to a large-scale database of neuronal responses and demonstrate the striking ability of GLIF models to both reproduce spiking behavior and classify a wide variety of transgenic lines. We show: (1) The traditional LIF model ($GLIF_1$) was able to reproduce 70% of the spike times within a resolution of 10 ms on an in vivo-like stimulus for a large set of biological neurons. Including additional mechanisms increased the ability of the models to recreate spike times by almost 10%. (2) Inhibitory neurons can be better fit than excitatory neurons, and we present evidence that inhibitory neurons are more stereotypical than excitatory neurons. (3) Inhibitory and excitatory models require different mechanisms to achieve the spike trains of neurons. (4) Increasing model complexity by adding mechanisms fit from the voltage waveform does not necessarily lead to increased performance in spiking behavior. Furthermore, mechanisms that do not increase performance independently can provide additional benefit when combined with other mechanisms. (5) Parameters obtained from fitting neurons with GLIF models are useful in classifying cell types: higher level GLIF parameters are more effective at differentiating cell types associated with transgenic lines than sub-threshold electrophysiological features. (6) GLIF parameters can be combined with spike-shape features for cell type classification.

We provide a large-scale database of point neuron models for 16 transgenic lines. We characterize the parameters of the GLIF models associated with the different transgenic lines and illustrate the neurons that best recreate the actual spike trains for each of the transgenic lines.

## Results

**Model description.** In GLIF models, the mechanisms are separated by time scale: GLIF models aim to represent the slow linear sub-threshold behavior of a neuron and recreate the spike times, not the shape of the action potential caused by fast, non-linear ion channels. Thus, none of the fast, non-linear processes associated with the action potential itself are included directly in the dynamics. However, some attributes of the spike, such as spike width and voltage after a spike relative to voltage before a spike, are accounted for in the reset rules which map the state before the spike to the state thereafter (Supplementary Methods "Parameter fitting and distributions" and Supplementary Fig. 1). Model

**Table 1 Description of model parameters and variables**

**Variables**

| Mechanism | Symbol | Variable | | |
|---|---|---|---|---|
| | $V(t)$ | Membrane potential | | |
| ASC | $I_j(t)$ | After-spike currents | | |
| reset | $\Theta_s(t)$ | Spike-dependent threshold component | | |
| adapt. th. | $\Theta_v(t)$ | Voltage-dependent threshold component | | |

**Parameters**

| Mechanism | Symbol | Parameter | Fit from | Post hoc opt |
|---|---|---|---|---|
| | $C$ | Capacitance | Sub-threshold noise | No |
| | $R$ | Membrane resistance | Sub-threshold noise | No |
| | $E_L$ | Resting potential | Resting $V$ before noise | No |
| | $\Theta_\infty$ | Instantaneous threshold | Short square input | Yes |
| | $\delta t$ | Spike cut length | All noise spikes | No |
| reset | $f_v$ | Voltage fraction following spike | All noise spikes | No |
| reset | $\delta V$ | Voltage addition following spike | All noise spikes | No |
| reset | $b_s$ | Spike-induced threshold time constant | Triple short square | No |
| reset | $\delta\Theta_s$ | Threshold addition following spike | Triple short square | No |
| ASC | $\delta I_j$ | After-spike current amplitudes | Supra-threshold noise | No |
| ASC | $k_j$ | After-spike current time constants | Supra-threshold noise | No |
| ASC | $f_v$ | Current fraction following spike | Set to 1 | No |
| adapt. th. | $a_v$ | Adaptation index of threshold | Prespike $V$ supra-thr. noise | No |
| adapt. th. | $b_v$ | Voltage-induced threshold time constant | Prespike $V$ supra-thr. noise | No |

equations can be found in the "Methods" section of the main text and the Supplementary Methods. Table 1 summarizes the model parameters and variables.

The family of GLIF models is schematized in Fig. 1a. Examples of model behavior are shown in Fig. 1b. A standard LIF model was our starting point, progressing to more generalized leaky integrate-and-fire models. In the standard LIF model ($GLIF_1$ here), current injected into the cell causes the voltage to rise in a linear fashion. When the voltage reaches a fixed threshold (referred to as $\Theta_\infty$ here) the model spikes and the threshold is reset to the resting potential of the neuron.

The $GLIF_2$ model advances the $GLIF_1$ model by incorporating more realistic reset rules ($R$) for voltage and threshold. The rapid changes of the action potentials are followed by slower dynamics which affect a neuron's state. The voltage after a spike does not reset to rest and the threshold does not remain at a fixed value. This $GLIF_2$ model continues to assume that the spikes are sufficiently similar such that a mapping between the voltage and threshold state before and after a spike can be found. The specific linear relationship of the voltage after the spike as a function of the voltage before the spike is found directly from the electrophysiological data (Supplementary Fig. 1). This relationship also defines the width of a spike or the "refractory period" which is implemented as a time in which the model cannot produce another spike. In addition to the refractory period, it is often more difficult to cause a neuron to spike again after a first spike due to mechanisms such as the slow inactivation of voltage-dependent currents which activate during the spike. This difficulty in causing another spike is modeled as a rise in threshold that decays after the spike and is again extracted directly from the data (Supplementary Fig. 3).

Although GLIF models do not aim to recreate the shape of an action potential caused by fast non-linear ion channel activation, the $GLIF_3$ model does incorporate the longer term effects of ion channels. Here we assume that ion channel currents have a stereotyped activation following a spike, and we bundle all slow ion-channel effects into a set of two after-spike currents (ASC) with different time scales. These currents are again fit directly from the data (see "Parameter fitting and distributions" in the Supplementary Methods).

The $GLIF_4$ model incorporates both reset rules of $GLIF_2$ and the after-spike currents of $GLIF_3$.

Finally, slow depolarization can lead to partial inactivation of the voltage-dependent sodium current which generates a spike. In $GLIF_5$ we incorporate this mechanism into an adapting threshold (AT) which is dependent on the membrane potential (see "Parameter descriptions" in the Supplementary Methods).

Note that the number of variables increases from one ($V(t)$) for $GLIF_1$ to two ($V(t), \Theta_s(t)$) for $GLIF_2$, etc., up to five for $GLIF_5$ (see Table 2).

**Data**. Intracellular electrophysiological recordings were carried out via a highly standardized process[24]. The data can be accessed in the Allen Cell Types Database[1]. The data considered here consists of in vitro electrophysiology data collected from 16 different labeled Cre-positive transgenic lines selective for different types of neurons. The transgenic line panel of Fig. 2 summarizes the properties of the different transgenic lines. Colors and short hand names represent all transgenic line data throughout this manuscript. In general, shades of red represent inhibitory transgenic lines and shades of blue represent excitatory lines. Excitatory neurons from each layer are identified using layer-selective lines. For inhibitory neurons, cells are targeted across lamina using lines selective for known neuronal subtypes[25]. Cells from each of the three major inhibitory subtypes (Pvalb (primarily basket cells), Sst (primarily Martinotti cells), and Htr3a (diverse morphologies)) are represented. Also included are transgenic lines that enrich for subsets of inhibitory types: the Vip, Ndnf, and Chat Cre lines (subtypes of Htr3a), Chrna2 (subtype of Sst), and Nkx2.1 (subtype of Pvalb). All cell types recorded are from young adult (P45 to P70) C57BL/6J mouse primary visual cortex.

**Model performance**. After model fitting and optimization (see Supplementary Methods for details; see Fig. 3 for distributions of the fit parameters) on our set of training stimuli (Fig. 2; see the "Stimulus" section of the Methods for details), we test model performance on a "hold out" in vivo-like noise stimulus not used for training (to ensure our models were not over fit). Using hold-out data to evaluate model performance is the "gold standard" for model selection. In the "Akaike Information Criterion" section of the Supplementary Methods and

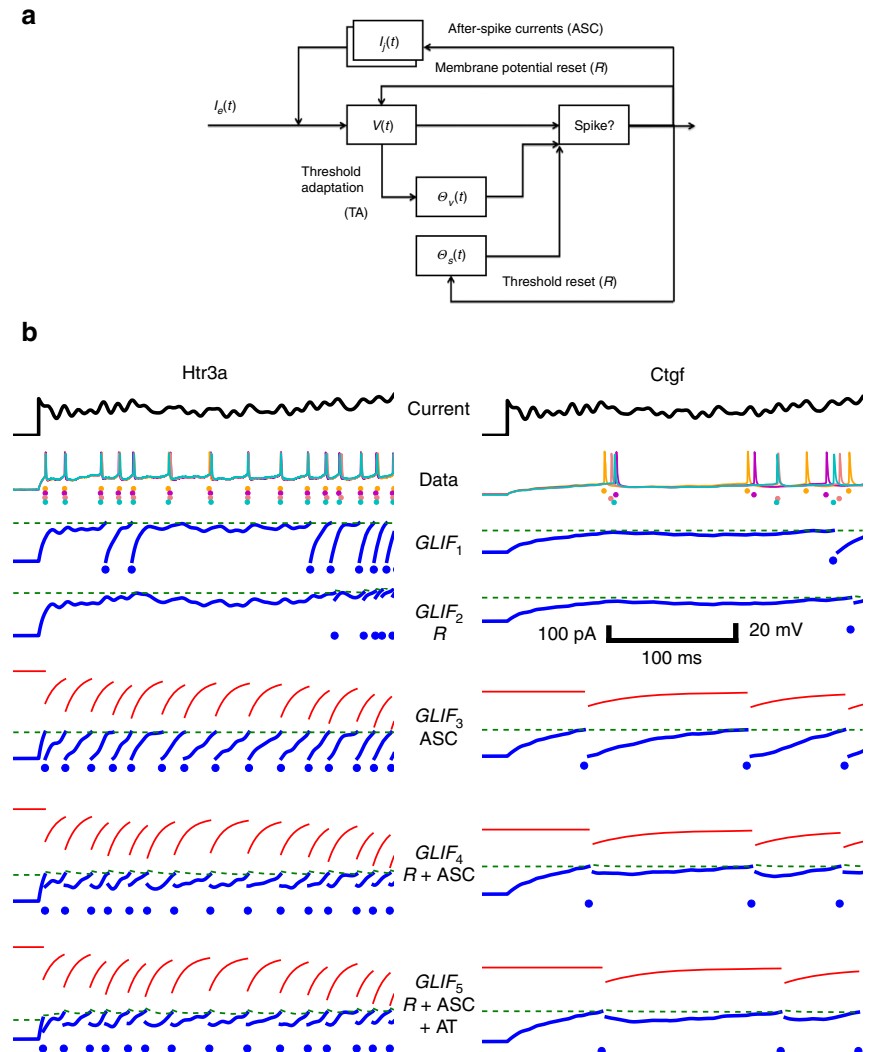

**Fig. 1** Five generalized leaky integrate-and-fire (GLIF) models consisting of different phenomenological mechanisms are fit to electrophysiological data. A schematic describing the mechanisms is shown in **a**. Example data and models from two neurons of different transgenic lines are shown in **b**. For all models the input is a current, $I_e(t)$, injected via a patch electrode illustrated in black at the top of **b**. Below the current are the voltage traces from four repeats of the same stimulus (here colors represent the different responses to the repeated stimuli and do not adhere to the standard color scheme in the rest of the manuscript). Below the biological data, the GLIF models are plotted. The output of the models is the trans-membrane potential, $V(t)$, pictured in blue. When $V(t)$ reaches a threshold, $\Theta = \Theta_\infty + \Theta_S(t) + \Theta_v(t)$, shown in dashed green, a spike is produced, illustrated by blue dots. Note that the shape of the spike is not plotted as it is not fit by these models. Instead, after a refractory period, $V(t)$ is reset to a value dependent on the specific model. The $GLIF_1$ model is equivalent to the traditional LIF model with a refractory period where the model can not spike. This model contains one variable, $V(t)$, and the threshold is fixed to a value we refer to as $\Theta_\infty$. $GLIF_2$ models include a second variable: a spike-induced threshold $\Theta_S(t)$ which is added to the baseline threshold $\Theta_\infty$. When the model spikes, $\Theta_S(t)$, jumps up and then decays. Thus, after a spike, initially the total threshold is higher making it harder for the model to reach threshold. $GLIF_3$ includes $V(t)$ and two variables corresponding to two spike initiated after spike currents, $I_j(t)$, which have different time constants and decay back to zero. The sum of the after-spike currents are illustrated in red. $GLIF_4$ combines $GLIF_2$ and $GLIF_3$ for a total of four variables. $GLIF_5$ includes an additional threshold component $\Theta_v(t)$. $\Theta_V(t)$ is dependent on the voltage of the model. Scale bars represent all model plots (not the amplitude of the current injection or biological voltage traces)

Supplementary Fig. 11, we also provide an Akaike Information Criterion (AIC) measure (which follows the same trends) to characterize the trade-off between complexity and model performance of GLIF models on training data. We quantified performance by how much of the temporal variance in the spike times of the data can be explained by the model at a 10 ms temporal scale (see Fig. 4 and "Evaluation of model spike times" in the Supplementary Methods for details). Medians of explained variance of all data can be viewed in Fig. 5 with corresponding values in Table 3. We begin our analysis by testing if there are overall differences between the GLIF model levels. A Friedman test reports that there are overall differences with a *p*-value of

1.83e−45. Thus, we continue with Wilcoxon sign-ranked tests where *p*-values are corrected for multiple comparisons using the Bejamini–Hochberg procedure. All *p*-values can viewed in Supplementary Figs. 8, 9, and 10.

The progression of explained variance through the different model levels demonstrates that different mechanisms are important for achieving spiking behavior of inhibitory and excitatory neurons. Overall $GLIF_1$ (equivalent to a leaky integrate and fire model) had a surprisingly high explained variance of 70% when all neurons were considered. Inhibitory neuron models perform better then excitatory models: 75% versus 68%. The introduction of reset rules in $GLIF_2$ decreases the performance of

**Table 2 Summary of GLIF models and results**

| | Num. cells | Variables | Model parameters | Parameters in clustering | Explained variance $\Delta t = 10$ ms | Num. clusters |
|---|---|---|---|---|---|---|
| GLIF$_1$ | 645 | $V(t)$ | $R, C, E_L, \Theta_\infty, \delta t$ | $R, C, E_L, \Theta_\infty, \delta t$ | 70.2% | 10 |
| GLIF$_2$ | 254 | $V(t), \Theta_s(t)$ | $R, C, E_L, \Theta_\infty, \delta t, f_v, \delta V, b_s, \delta\Theta_s$ | $R, C, E_L, f_v, \delta V, \Theta_\infty, \delta t$ | 67.7% | 15 |
| GLIF$_3$ | 645 | $V(t), I_1(t), I_2(t)$ | $R_{ASC}, C, E_L, \Theta_\infty, \delta t, k_1, \delta I_1, k_2, \delta I_2$ | $R_{ASC}, C, E_L, \Theta_\infty, \delta t, \delta I_1/k_1, \delta I_2/k_2$ | 72.4% | 18 |
| GLIF$_4$ | 254 | $V(t), \Theta_s(t) I_1(t), I_2(t)$ | $R_{ASC}, C, E_L, \Theta_\infty, \delta t, f_v, \delta V, b_s, \delta\Theta_s, k_1, \delta I_1, k_2, \delta I_2$ | $R_{ASC}, C, E_L, \Theta_\infty, \delta t, f_v, \delta V, \delta I_1/k_1, \delta I_2/k_2$ | 75.9% | 16 |
| GLIF$_5$ | 253 | $V(t), \Theta_s(t) I_1(t), I_2(t), \Theta_v(t)$ | $R_{ASC}, C, E_L, \Theta_\infty, \delta t, f_v, \delta V, b_s, \delta\Theta_s, k_1, \delta I_1, k_2, \delta I_2, a_v, b_v$ | N/A: additional parameters are not available for all 645 neurons. | 77.6% | N/A |
| All features | 645 | N/A | N/A | $\tau_m, R_i, V_{rest}, I_{thresh}[†], V_{thresh}[†], V_{peak}[†], V_{fasttrough}[†], V_{trough}[†]$ up:downstroke[†], up: downstroke[*], sag, $f$-$I$ curve slope, latency, max. burst index | N/A | 16 |
| Sub-thr features | 645 | N/A | N/A | $\tau_m, R_i, V_{rest}, I_{thresh}[†], V_{thresh}[†], V_{trough}[†]$, sag, $f$-$I$ curve slope, latency, max. burst index | N/A | 16 |

The "Num. cells" column reports the number of cells for which a model was constructed or the paradigm was clustered. Note that for the GLIF models clustering was only performed on parameters that were available for all 645 cells. Therefore, the "Parameters in clustering" list is a subset of the total "Model Parameters" available for any level. The variables for each model level are listed in the "Variables" column. Note that resistance was fit along with after-spike currents in models where after-spike currents were implemented. $R$ denotes the resistance fit without ASC and $R_{ASC}$ denotes the resistance fit along with after-spike currents. GLIF$_5$ does not report clusters because there are no additional parameters available for all 645 neurons, i.e., it would be the same clustering paradigm as in GLIF$_4$ as the only new parameters associated with GLIF$_5$ ($a_v$ and $b_v$) are only fit for the reduced set of cells. We are unable to cluster on the time scales of every after-spike current alone as there were five discrete possible values but only two were chosen for each neuron. Therefore, we cluster on the total charge deposited over short $\delta I_1/k_1$ and long $\delta I_2/k_2$ time scales (continuous numbers) for the model levels that contain after-spike currents. The average explained variance at a time resolution of 10 ms for all neurons at each level is reported as well as the number of clusters that were found using the aforementioned clustering parameters via the hierarchical clustering technique. As in Supplementary Tables 6 and 7, * denotes features measured during a short square stimulus, and † represents features measured during a long square stimulus (Fig. 2)

the models for both excitatory (65%) and inhibitory (74%) neurons. The introduction of after-spike currents alone in GLIF$_3$ helps the inhibitory neurons (81%) but not the excitatory (68%) neurons. To confirm that improvement of fitting between GLIF$_1$ and GLIF$_3$ follow different trends for inhibitory and excitatory neurons, we subtracted the pairwise explained variance ratio of GLIF$_1$ from GLIF$_3$ models. We found that the distributions of these differences for the excitatory and inhibitory neurons are significantly different ($p = 1.35e-7$, Mann–Whitney $U$; Benjamini–Hochberg correction, family size = 10), suggesting that after-spike currents have different influence in inhibitory and excitatory neurons. Interestingly, the inclusion of reset rules with after-spike currents in GLIF$_4$ improves the performance of both excitatory (72%) and inhibitory neurons (83%) even though reset rules alone hurt performance. Please see the "Discussion" section for a discourse on the influence of model complexity on performance. Here again, the distribution of differences between GLIF$_3$ and GLIF$_4$ are statistically different for inhibitory and excitatory neurons ($p = 2.89e-4$, Mann–Whitney $U$; Benjamini–Hochberg correction, family size = 10). Finally, the addition of the adaptive threshold along with the after-spike currents and reset rules provides an improvement for excitatory cells (75%) but does little for inhibitory (84%) cells. Distributions of differences of excitatory and inhibitory neurons differ with the following values GLIF$_3$ and GLIF$_5$: $p = 1.98e-5$, GLIF$_4$ and GLIF$_5$: $p = 1.40e-2$ (Mann–Whitney $U$; Benjamini–Hochberg correction, family size = 10).

We found it interesting that the addition of reset rules measured directly from the data actually hindered the ability of the GLIF$_2$ model to recreate spike times of the neural data. We explored if there was a relationship between the ability of a model to reproduce the sub-threshold behavior of the voltage waveform and reproduce the neural spike times. When taking into account all models, overall there is a correlation between the ability of a model to reproduce the sub-threshold voltage and its ability to reproduce spike times (please see the black regression line in Supplementary Fig. 15). However, when considering the median performance values describing the ability of the different model levels to reproduce sub-threshold voltage and spike times

(Supplementary Fig. 15), although GLIF$_2$ better reproduces sub-threshold voltage than GLIF$_1$ and GLIF$_3$, it does worse at reproducing the spiking behavior. Thus, the ability of a model to reproduce sub-threshold voltage does not necessarily translate into better spike time performance.

There are several potential methodological and biological explanations for the increased ability to fit inhibitory neurons over excitatory neurons. It is possible that inhibitory neurons are better optimized because they generally fire more and thus have more data points to optimize (see the Supplementary Methods for details). Alternatively, perhaps they are easier to fit because they are more stereotypical than excitatory neurons. As described in the "Parameter fitting and distributions" section of the Supplementary Methods and Supplementary Fig. 1, we define spike cut length as the post spike time at which a linear regression based on the voltage before the spike minimizes the squared residuals of the voltage after the spike. We use the standard error from this spike cut length fit as a description of spike reproducibility. Inhibitory neurons have shorter spike cut lengths than excitatory neurons (median 2.7 versus 4.6 ms, respectively; $p = 1.10e-47$, Mann Whitney U), spike more frequently (a median of 84 versus 38 spikes per noise 1 stimuli, respectively; $p = 6.15e-35$, Mann Whitney U), and have more reproducible spikes (median standard error $2.6e-2$ and $5.7e-2$, respectively; $p = 4.97e-39$, Mann Whitney U). Supplementary Fig. 4 shows there are correlations between the explained variance ratio for each of these three variables. To investigate the importance of these factors in achieving a good model fit, we perform a multiple linear regression (Python, statsmodels.regression.linear_model.OLS). Spike cut length and number of spikes are good indicators of how well a neuron will be fit ($p$-values are $2.80e-19$ and $5.95e-21$, respectively), whereas this measure of spike reproducibility is not statistically important ($p$-value: 0.276).

It is likely that individual transgenic lines will have different mechanisms most important for achieving their spiking behavior. We do not draw any conclusions here due to a large number of non-significant $p$-values between model levels for individual transgenic lines (most likely due to the lower number of neurons from individual transgenic lines in this study). Performance of

Transgenic lines

| Name | n | E/I | L1 | L2/3 | L4 | L5 | L6 |
|---|---|---|---|---|---|---|---|
| All neurons | 645 | E/I | X | X | X | X | X |
| Inhibitory | 283 | I | X | X | X | X | X |
| Excitatory | 362 | E | | X | X | X | X |
| Scnn1a-Tg2 | 22 | E | | | X | | |
| Scnn1a-Tg3 | 37 | E | | | X | X | |
| Nr5a1 | 47 | E | | | X | | |
| Rorb | 88 | E | | | X | X | |
| Cux2 | 56 | E | | X | X | | |
| Ntsr1 | 39 | E | | | | | X |
| Ctgf | 20 | E | | | | | X |
| Rbp4 | 53 | E | | | | X | |
| Sst | 64 | I | | X | X | X | X |
| Pvalb | 53 | I | | X | X | X | X |
| Htr3a | 62 | I | X | X | X | X | |
| Ndnf | 15 | I | X | | | | |
| Chat | 18 | I | | X | X | X | X |
| Vip | 19 | I | | X | X | X | X |
| Chrna2 | 29 | E/I | | | X | X | |
| Nkx2.1 | 23 | I | | X | X | X | X |

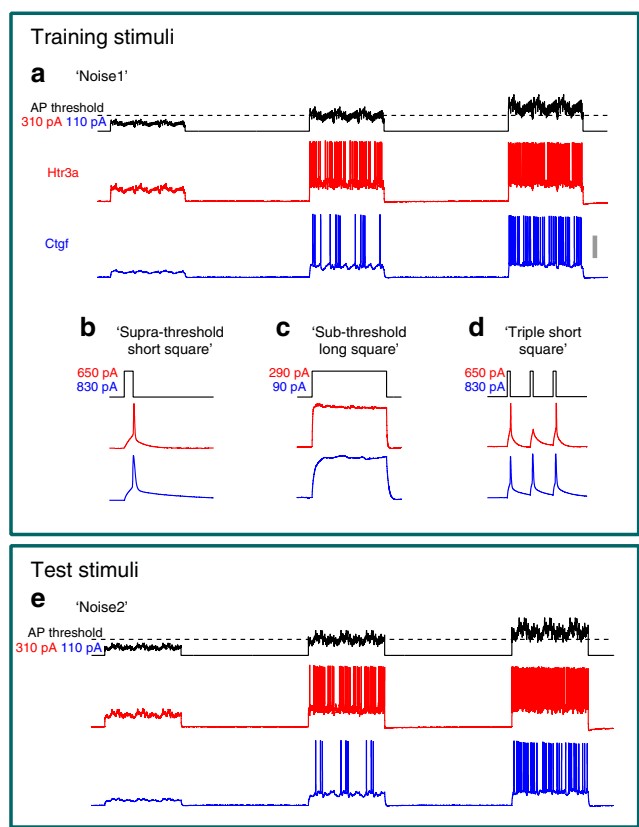

**Fig. 2** Overall, 645 different neurons from 16 transgenic lines containing all the required stimuli on the Allen Cell Types Database are considered in this study. (Left) Illustrated colors correspond to the different transgenic lines in all figures. "*n*" describes the number of neurons for which the lowest level model ($GLIF_1$) could be generated. Transgenic lines can identify either or both inhibitory (I) or excitatory (E) cells, which reside in layer 1 (L1) through layer 6 (L6). Note that most Chrna2-Cre-positive neurons are inhibitory and thus are labeled here as inhibitory. (Right) A minimal set of stimuli were required for training and testing different GLIF models. GLIF models were trained using **a** at least two repeats of pink noise stimuli (3 s each, 1/*f* distribution of power, 1–100 Hz) with amplitudes centered at 75, 100, and 125 percent of action potential threshold, **b** a short (3 ms) just supra-threshold pulse to fit the instantaneous threshold, $\Theta_\infty$, **c** a long square (1 s) pulse just below threshold to estimate the intrinsic noise present in the voltage traces (used in the post hoc optimization step), and **d** a series of three peri-threshold short pulse sets for any model with reset rules ($GLIF_2$, $GLIF_4$, and $GLIF_5$). GLIF models were then tested using a hold-out stimulus set **e** of at least two sweeps of a second pink noise stimuli generated in an identical manner to the training but initialized with a different random seed. Representative data shown from an Htr3a-Cre-positive neuron (resting membrane potential (RMP) −67 mV) and a Ctgf-Cre-positive neuron (RMP = −74 mV). Scale bar in **a** corresponds to 40 mV (**a**, **e**), 50 mV (**b**, **d**), or 20 mV (**c**)

individual transgenic lines can be viewed in Fig. 5 with corresponding values in Table 3 and Supplementary Figs. 9 and 10.

**Example neurons**. Throughout the manuscript data and models from two exemplary neurons are consistently shown as examples: an Htr3a inhibitory neuron (specimen ID 474637203) and a Ctgf pyramidal neuron (specimen ID 512322162).

To provide examples of models that do well reproducing the spike times of the different transgenic cell lines, for each line we select the model which has the highest explained variance from neurons that have all five model levels. Example neurons are labeled with stars in all figures and can be found on the website via the following specimen IDs: Nr5a1: 469704261 $GLIF_5$, Cux2: 490376252 $GLIF_4$, Ctgf: 512322162 $GLIF_5$, Vip: 562535995 $GLIF_5$, Scnn1a-Tg2: 490205998 $GLIF_5$, Ntsr1: 490263438 $GLIF_4$, Sst-IRES: 313862134 $GLIF_4$, Rbp4: 488380827 $GLIF_5$, Scnn1a-Tg3-Cre: 323834998 $GLIF_3$, Chrna2: 580895033 $GLIF_2$, Pvalb: 477490421 $GLIF_4$, Ndnf: 569623233 $GLIF_5$, Nkx2.1: 581058351 $GLIF_5$, Htr3a: 474637203 $GLIF_5$, Chat: 518750800 $GLIF_5$, Rorb: 467003163 $GLIF_3$. Although these neurons are the best performers, they may not be the most representative cells. Most of these models have at least one parameter which lies outside the 5 to 95 percentiles for the trangenic line. To facilitate the choice of

other good representative models, neurons that contain all GLIF model levels and all model parameters within the 5 to 95 percentiles are available in Supplementary Table 1.

**Clustering**. After deriving parameters for each of the GLIF models, we assessed how well these parameters could classify neurons into putative types corresponding to transgenic lines. We used two different clustering algorithms to identify trends that were not method-specific. The first of these is an iterative binary splitting method, using standard hierarchical clustering methods (see the "Clustering" section of the Supplementary Methods for details) to separate cells into groups in an iterative manner. We show the results of the hierarchical clustering here in Figs. 6 and 7. We confirm the general trends from this analysis using a second clustering method, affinity propagation[26], as shown in Supplementary Figs. 12 and 14.

To compare different partitions of the cells, we use the adjusted Rand index (ARI)[27] and the adjusted variation of information (AVOI) metric[28]. The adjusted Rand index is a corrected-for-chance extension of the Rand Index, which measures the probability that any pair of elements will belong to the same cluster in different partitionings. A value of 1 indicates that the data clusterings are exactly the same, whereas a value of 0

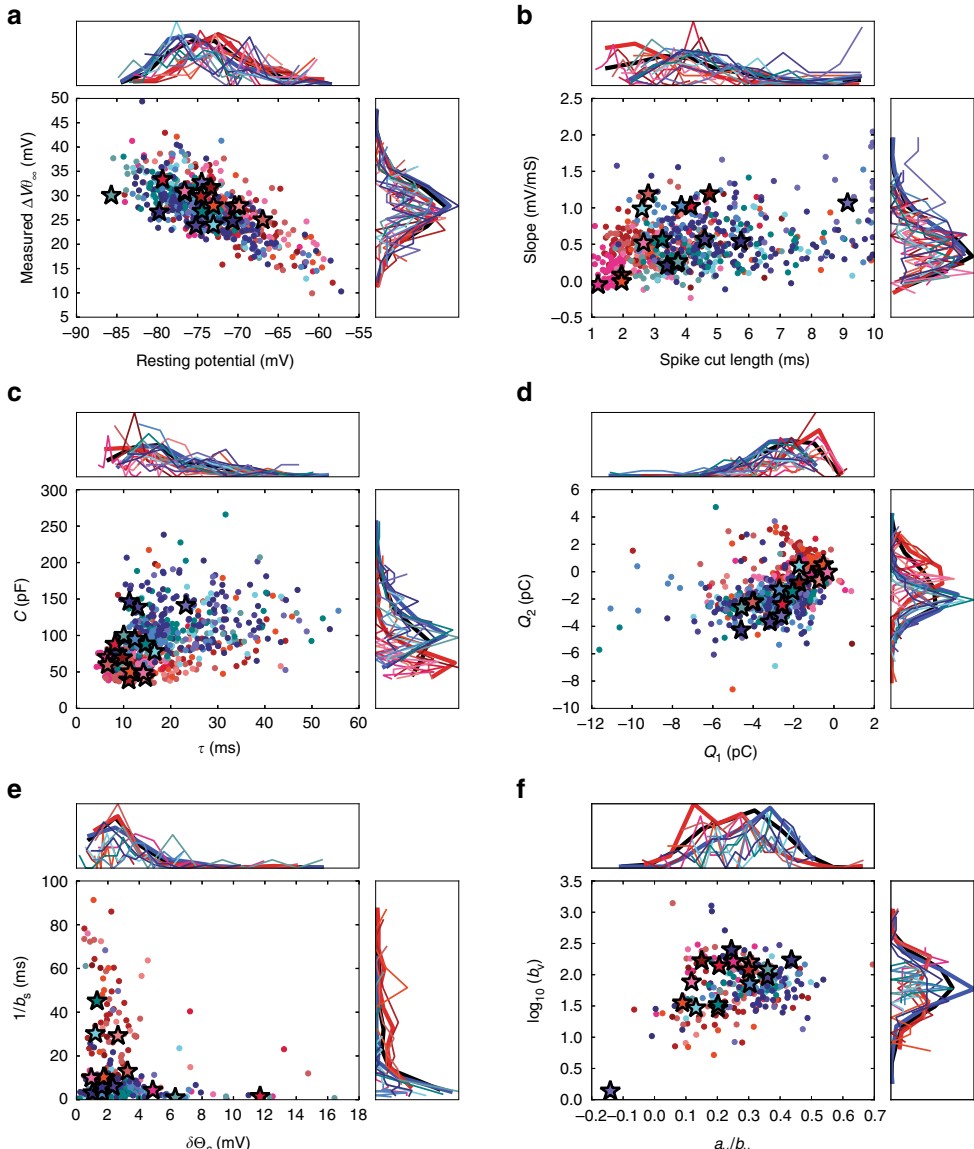

**Fig. 3** Slices from the parameter space fit from electrophysiological data. Detailed parameter fitting methodology (Fig. 2) is available in the Supplementary Methods. Stars denote example neurons listed in the main text. **a** Resting potential is measured as the average voltage during rest before training noise (noise 1) current is injected. The threshold relative to rest, $\Delta V$, is measured by subtracting resting voltage from the threshold obtained from the supra-threshold short square pulse. **b** The spike waveform is removed from the voltage trace by aligning all spikes and fitting a line to the voltage before and after a spike. The best fit line within a window of 10 ms after spike initiation was chosen (Supplementary Fig. 1). This spike cut length is used in all models, and voltage measurements before and after a spike are used to reset voltage in ($GLIF_2$, $GLIF_4$, and $GLIF_5$). **c** Capacitance and resistance are fit via linear regression to sub-threshold voltage (Supplementary Fig. 2). The membrane time constant, $\tau = RC$, is plotted. **d** Total charges of the fast, $Q_1$, and slow, $Q_2$, after-spike currents deposited each time there is a spike. **e** The amplitude, $a_s$, and decay, $b_s$, of the spiking component of the threshold, $\Delta_s$, (used in $GLIF_2$, $GLIF_4$, and $GLIF_5$) is fit to the triple short square data set (Supplementary Fig. 3). **f** In the $GLIF_5$, the threshold is influenced by the voltage of the neuron according to (Equation 4). The two parameters of Eq. 4 are plotted here. Colors in all panels correspond to transgenic lines illustrated in Fig. 2

indicates that the clusterings are no different from what would be expected by chance. The AVOI metric is an information-theoretic value measuring the ability of one partitioning to predict the other, as compared to shuffled-label data. A value of 0 indicates that the clusterings are no different from what would be expected by chance. The upper bound on the AVOI is dependent on the number of elements in the clustering. Here the upper bound is 6.47. We show the ARI and AVOI from the hierarchical clustering in the main text (Figs. 6 and 7). We show results from affinity propagation in the Supplementary Figs. 12 and 14. The variability in the performance of the binary splitting method via bootstrapping is shown in Supplementary Fig. 17. As mentioned

in the "Stimulus" section of the Methods, $GLIF_2$, $GLIF_4$, and $GLIF_5$ require a stimulus that is not applied to all neurons (for reasons independent of this study). To ensure a full data set for clustering, we use only parameters that are available for all 645 cells. Table 2 describes the parameters used in each clustering paradigm.

First, we assess how well the putative clusters agree with known cell type information based on the transgenic lines from which the cells were derived. This is an imperfect validation as it is known that different transgenic lines may not perfectly correspond to different cell types, as measured by transcriptomics[25]. The ARI and AVOI values for Cre lines compared to

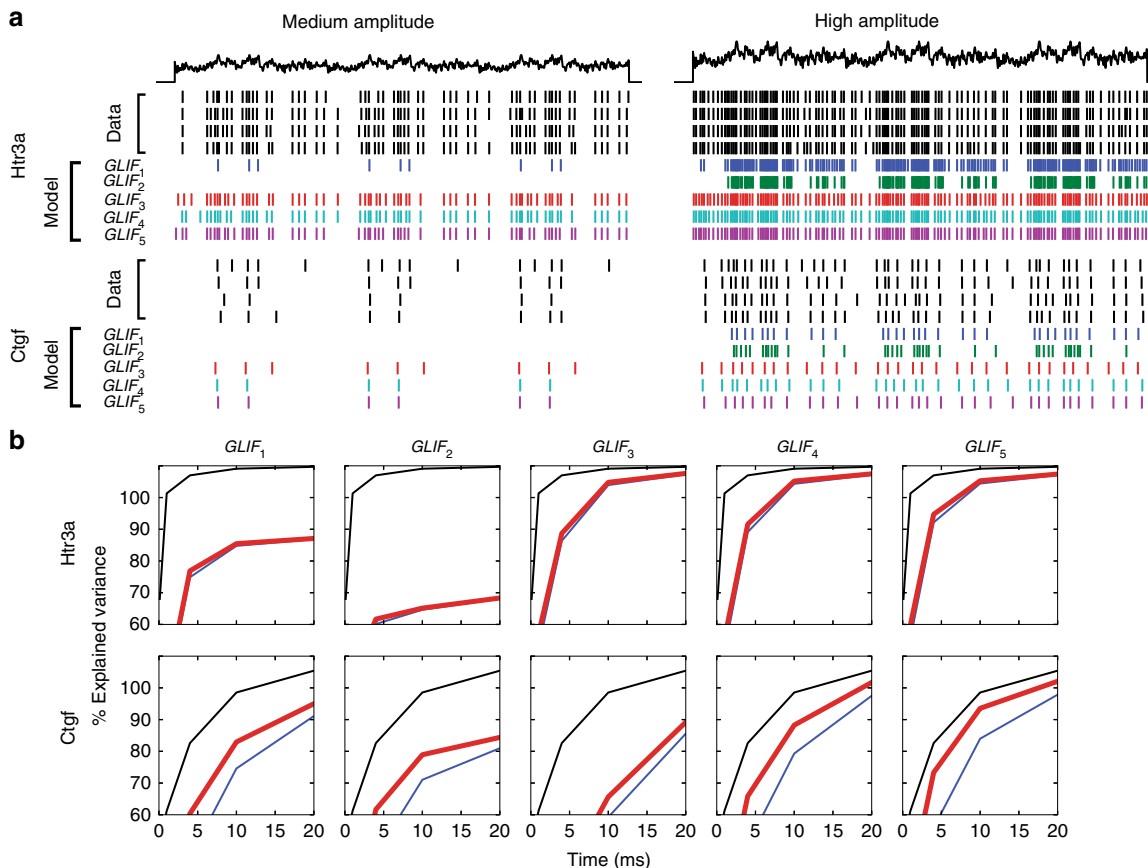

**Fig. 4** Rastergrams and explained variance of biological data and all optimized model levels for "hold out" test data. **a** Data for the two example cells. Injected current shown in black. Black rasters are spikes from recorded neurons to repeated current injections. Colored rasters correspond to the five different, deterministic models. The current injection is 3 s long. As $GLIF_1$ and $GLIF_2$ do not have a spike frequency adaptation mechanism (implemented by the summation of after-spike currents over many spikes in $GLIF_3$), they have trouble reproducing simultaneously the firing patterns at multiple input amplitudes. **b** Explained variance for the different model levels at different levels of time window resolution $\Delta t$. The black lines represent the explained variance of the data (how well the neuron repeats its own spiking behavior). This trace would reach 100% if the spike times of the data were all exactly at the same time in each repeated stimulus: the fact that they are not 100% reflects the intrinsic variation in the spike times within the experimental data. The blue line illustrates the pairwise explained variance of the model with the data. Because the model can not be expected to explain the data better than the data can explain itself, the red line is the ratio of the pairwise explained variance of the model (blue) and the data divided by the explained variance of the data (black). This ratio value at a $\Delta t = 10$ ms time bin is used for the explained variance performance metric in the main text

transcriptomics found in a previous study[25], are 0.30 and 2.69, respectively (AVOI upper bound was 6.86 in the cited study). Nonetheless, we would expect some degree of relationship, as broad classes of cells do tend to segregate by transgenic line. Second, to further validate our results, we compare the clustering obtained by GLIF parameters to the clustering obtained using features extracted directly from electrophysiological traces. Here we use two sets of features. In the first set, we cluster on 14 features that contain aspects of both sub-threshold and supra-threshold spike-shape features. As GLIF neurons do not recreate the shape of a spike, in the second set we start with the same set of features but eliminate all features that describe the shape of the action potential. See the "Electrophysiological features" section for a brief description of the features used here; a more detailed description can be found in ref. [24]

The progression of the ability of the GLIF model parameters to recapitulate segregation of cells by transgenic line is shown in Fig. 6. Figure 6 quantifies how clustering the different GLIF level parameters improves the distinction among different transgenic lines. Although $GLIF_1$ and $GLIF_2$ have an ARI of ~0.08 (AVOI ~0.8), $GLIF_3$ ARI increases to ~0.12 (AVOI ~1.3) and $GLIF_4$ improves even more yielding a value of ~0.18 for the iterative binary clustering (AVOI ~1.3). Similarly, the ARI and AVOI

values calculated during affinity propagation are higher for $GLIF_3$ and $GLIF_4$ than for $GLIF_1$ and $GLIF_2$ (Supplementary Fig. 12).

The transgenic line composition of each terminal cluster in $GLIF_4$ (Fig. 6) suggests that clustering based on model parameters broadly segregates neurons into previously identified classes of cells. Neurons from transgenic lines labeling predominantly excitatory neurons cluster separately from those labeling mainly interneurons. In addition, sub-categories of neurons appear: these include parvalbumin-positive interneurons, layer 6a corticotha-lamic neurons (derived from the Ntsr1 transgenic line), and layer 6b neurons (derived from the Ctgf transgenic line).

When the same hierarchical clustering technique is applied directly to the electrophysiological feature data set that includes spike-shape parameters (bottom left of Fig. 6 and Supplementary Fig. 14), the correspondence between electrophysiological clusters and transgenic lines is similar to those of $GLIF_4$ parameter-based clusters (the electrophysiological cluster scores higher than $GLIF_4$ when using iterative binary clustering (Fig. 6) and lower when using affinity propagation (Supplementary Fig. 12). However, when the features describing the spike shape of the neuron are removed, the ARI drops below $GLIF_4$ to a value similar to $GLIF_3$ (Fig. 6). Thus, the dimensionality reduction onto the space of model fit parameters allows for discriminability better than that

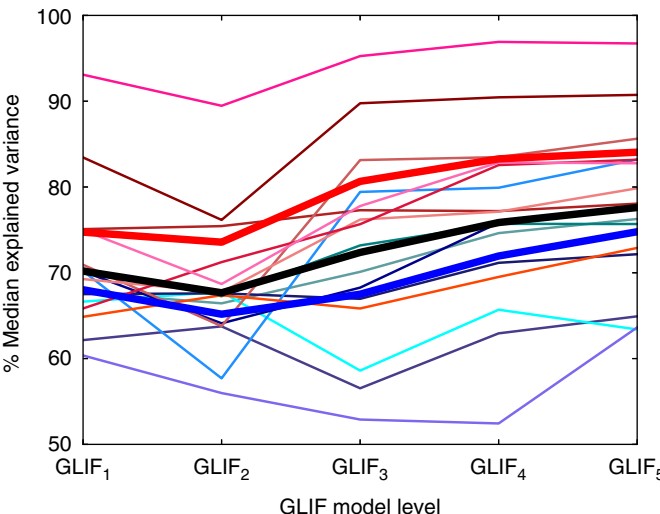

**Fig. 5** Different mechanisms improve model performance for inhibitory and excitatory neurons. The traditional leaky integrate and fire model ($GLIF_1$) yielded surprisingly high model performance. Overall, inhibitory models were more successful at reproducing spike times than excitatory models. Reset rules implemented on their own ($GLIF_2$) decreased model performance. After-spike currents ($GLIF_3$) improved inhibitory model performance, whereas a combination of both after spike currents and reset rules ($GLIF_4$) were required to gain performance of excitatory models. The voltage-dependent adapting threshold ($GLIF_5$) improved performance of excitatory models even more, but had only a slight effect on inhibitory models. The thick blue line denotes all excitatory neurons, the thick red line denotes all inhibitory neurons, and the thick black line is for all neurons. Thin lines are different transgenic lines. Accompanying data is available in Table 3. p-values for significant differences between GLIF model levels can be found in Supplementary Figures 8, 9, and 10. Briefly, for the "all", "excitatory", and "inhibitory" groupings the p-values are smaller than 0.01 (and often much smaller) for all but between the excitatory $GLIF_2$ and $GLIF_3$ (Supplementary Fig. 8). Differences between GLIF levels of different transgenic lines are sometimes statistically significant and sometimes not (Supplementary Figures 9 and 10)

obtained by the extraction of a subset of electrophysiological features without spike shape.

Similarly, when we compare how well the GLIF parameter clustering matches the full feature-based clustering (Fig. 7, bottom right), $GLIF_1$ and $GLIF_2$ parameter-based clustering results in partitions that are dissimilar from that obtained using the electrophysiological features. $GLIF_3$ and $GLIF_4$ parameter-based clustering results in partitions that are closer to that obtained with the full-feature clustering. The set of partitions obtained by the non-spike electrophysiological features are the most similar to the partitions obtained from the full set of electrophysiological features. This is not surprising as the non-spike features are a subset of the full set of features. Confusion matrices of the relationship between electrophysiological feature-based clustering and GLIF parameter-based clustering are shown in Supplementary Fig. 13.

We performed clustering on the GLIF parameters and the spike features to determine whether combining them would further help the differentiation of Cre lines (Fig. 7 and Supplementary Fig. 14). Including spike-shape features along with $GLIF_1$ and $GLIF_2$ parameters greatly increases their ability to differentiate Cre lines. For the more complex neuronal models, the improvement of including the spike features is modest, suggesting that the $GLIF_3$ and $GLIF_4$ model parameters encompass most of

the information contained in this set of sub and supra-threshold electrophysiological features.

## Discussion

In the cerebral cortex there is a wide diversity of neuron types observed at transcriptomic levels[25]. The existence of different types may be driven by a need to develop particular cell type-specific connectivity, neuromodulation, or perform different cell specific computations, as well as by developmental and evolutionary constraints. When constructing a system-level model, it is important to know how many types of neurons are needed.

We provide a sizable database of GLIF models that can facilitate the development of system models with cell types available in the Allen Cell Types Database at http://celltypes.brain-map.org/. Here we show that GLIF models can simultaneously reproduce the spike times of biological neurons and reduce the complex mapping between the input current to the spike-train output with a small number of parameters. We explore how complexity affects the ability of models to both reproduce spike times and differentiate transgenic lines via unsupervised clustering. We find that the more complex models ($GLIF_3$ or greater) are better than the less complex models at reproducing biological spike times and differentiating transgenic lines.

After optimizing the neuronal parameters, we were surprised at how well the traditional leaky integrate and fire neuron models reproduce spike times under naturalistic conditions, explaining a median value of 70.2% of the variance. To put this value in context, biophysically realistic models with passive dendrites in the Allen Cell Types Database[1] achieve a median explained variance of 65.1% ($n = 195$) and biophysically realistic models with active dendrites achieve a median explained variance of 69.3% ($n = 107$). It should be noted that the biophysical models are not optimized to reproduce spike times but instead to recreate other aspects of electrophysiological responses[8,29]. As we show with $GLIF_2$, fitting data to aspects of the voltage waveform does not mean that the ability of the models to reproduce spike times will improve.

Our result that a model's ability to reproduce sub-threshold or spiking behavior does not increase monotonically with model complexity may seem counterintuitive. Furthermore, we demonstrate that the overall ability of a model level to reproduce sub-threshold behavior does not translate into the overall ability of a model level to reproduce spike times. When trying to understand how this is possible, it is important to note several things. Theoretically, in a situation where all parameters are simultaneously optimized, increasing complexity (via the addition of new mechanisms and variables to the previous model) should lead to a smaller error of the function being minimized. In this paradigm, one could include more parameters or state variables until the model starts to over fit the training data. Optimizing on our performance metric of explained variance would no longer be convex, and hence, optimization would be more unintuitive and computationally intensive with no guarantee of convergence. We chose a different route, in which different observables and error functions are used to isolate the effects of individual mechanisms. For example, here we optimize the difference in voltage between the model and the data during the forced spike paradigm because this is a convex problem. What we desired was a "good" model that has mechanisms which relate to how the neuron is actually performing computations and thus recreates observed behavioral aspects such as sub-threshold voltage and spike times. In our study, the following procedures could lead to the observed non-monotonic behavior. (1) Individual mechanisms are not optimized to reproduce the spike times, but rather to fit different aspects of the sub-threshold membrane potential during different

**Table 3 Explained variance of all model levels after post hoc optimization**

|  | n | GLIF$_1$ | GLIF$_2$ | GLIF$_3$ | GLIF$_4$ | GLIF$_5$ |
|---|---|---|---|---|---|---|
| All | 645 | 70.2 (43.8, 93.7) | 67.7 (47.4, 90.5) | 72.4 (40.7, 95.3) | 75.9 (46.8, 96.5) | 77.6 (49.4, 96.5) |
| Inhibitory | 283 | 74.7 (41.7, 95.6) | 73.6 (45.5, 93.8) | 80.7 (42.6, 97.2) | 83.3 (52.1, 97.5) | 84.1 (52.3, 97.3) |
| Excitatory | 362 | 68.0 (47, 81.6) | 65.2 (49.5, 75.2) | 67.5 (39.8, 86.9) | 72 (46.2, 88) | 74.8 (46.6, 89.1) |
| Scnn1a-Tg2 | 22 | 62.2 (39.6, 74.4) | 63.8 (52.3, 70.5) | 56.5 (36, 74.8) | 62.9 (38, 72.8) | 64.9 (37.5, 75.3) |
| Nr5a1 | 47 | 67.7 (50.8, 80.1) | 66.4 (61.6, 76.3) | 70.1 (46.3, 85.7) | 74.6 (49.4, 85.6) | 76.3 (57, 86) |
| Scnn1a-Tg3 | 37 | 66.6 (49.7, 79) | 67.8 (56.4, 80.1) | 58.6 (42.2, 89) | 65.7 (39.9, 88.5) | 63.4 (48.2, 88.6) |
| Rorb | 88 | 67.5 (46.4, 81.1) | 67.6 (56.9, 77.2) | 67 (39.1, 86.5) | 71.2 (51.3, 85.2) | 72.2 (50.7, 87) |
| Cux2 | 56 | 70.5 (59.1, 82.2) | 67.3 (53.4, 75) | 73.2 (42.9, 86.3) | 75.7 (56.1, 87.7) | 75.7 (60, 87.6) |
| Ntsr1 | 39 | 70.2 (52.1, 80.4) | 57.7 (38.4, 68.7) | 79.4 (55.4, 88.4) | 79.9 (69.2, 91.1) | 83.2 (69.2, 91.6) |
| Ctgf | 20 | 60.4 (34.9, 74.2) | 56.0 (44.2, 66) | 52.9 (33.5, 78) | 52.4 (38, 78.4) | 63.6 (33.7, 81.3) |
| Rbp4 | 53 | 70.2 (49.6, 85.1) | 64.1 (52.5, 73.8) | 68.3 (43.5, 86.5) | 75.8 (53.8, 87.2) | 77.5 (58.2, 88.1) |
| Sst | 64 | 75.1 (32.2, 93.5) | 75.4 (46, 90.9) | 77.3 (37, 95.8) | 77.2 (51.7, 95.8) | 78.1 (49.3, 95.7) |
| Pvalb | 53 | 93.1 (32.4, 96.9) | 89.5 (43.4, 96.3) | 95.3 (34.6, 97.6) | 96.9 (53.2, 98.5) | 96.7 (54.9, 98.6) |
| Htr3a | 62 | 70.9 (49.6, 83.2) | 63.8 (45.5, 80) | 83.1 (49.2, 94) | 83.5 (57.7, 94) | 85.6 (64.7, 93.9) |
| Ndnf | 15 | 83.4 (59.3, 93) | 76.2 (55, 83.1) | 89.8 (58.2, 92.6) | 90.5 (60, 94.4) | 90.7 (65.9, 94.2) |
| Chat | 18 | 65.8 (44.2, 80.7) | NaN | 75.7 (43.6, 85.3) | NaN | NaN |
| Vip | 19 | 69.3 (41.7, 85.3) | 67.7 (50.8, 72.7) | 76.2 (52.6, 89.3) | 77.1 (62.8, 86.6) | 79.8 (63.5, 86.7) |
| Chrna2 | 29 | 64.9 (52.6, 89.3) | 67.4 (51.2, 88.2) | 65.8 (43.4, 91) | 69.5 (53.5, 95.6) | 72.9 (54.5, 95.9) |
| Nkx2 | 23 | 75.0 (59.6, 97.4) | 68.7 (60.4, 96.2) | 77.8 (58.1, 95.3) | 82.9 (68.2, 97.5) | 82.8 (68.4, 97.5) |

The single number in each cell is the median. The first and third quartiles are in brackets below the median. All Cre lines with five or more neurons (*n*) present in the *GLIF*$_1$ level are included. Some neurons have the required stimuli for LIF models but do not have the stimuli required for higher level models. When there are not more than five neurons in the level, the values are denoted with a NaN. Data illustrated in Supplementary Figures 8, 9, and 10

stimuli. (2) The biological effects phenomenologically described here by reset rules and the after-spike currents act together to create the voltage waveform of a neuron. Here, we fit the after-spike currents and the reset rules independently on the voltage waveform of a neuron without subtracting out the effect of the other mechanism. Explicitly, when the reset rules in GLIF$_2$ were calculated, there were certainly after-spike currents (which loosely represent the summation of the ion-channel effects in biological neurons) present in the data. In addition, the after-spike currents were fit to the observed sub-threshold voltage, which includes the biological reset rules. (3) Reset rules and after-spike currents exert influence on different time scales. Reset rules have an instantaneous effect implemented right after a spike (or far away from the next spike) whereas after-spike currents have multiple time scales that extend though the sub-threshold portion of the voltage trace to the next spike. (4) We only optimize one parameter to the spike times of the data. More parameters could be optimized to potentially achieve better spike time performance. However, adding more parameters greatly increases optimization time and is more prone to over-fitting. (5) The metrics we used to fit and optimize the parameters (linear regression and difference between model and data voltage at spike time in the forced spike paradigm) were not our reporting metrics (explained variance, difference in sub-threshold voltage). The "take home" message here is that the ability to accurately reproduce spike times is due to the complicated interaction of many counteracting mechanisms over different time scales.

The progression of the explained variance for the different GLIF levels suggests that after-spike currents are very important for achieving the spiking behavior of inhibitory neurons, whereas a combination of after-spike currents and reset rules are necessary to improve the performance of excitatory neurons. In addition, incorporating the voltage-activated threshold improves the performance of excitatory neurons even more while it does little for inhibitory models. At all levels, models were able to reproduce the spike times of inhibitory neurons better than excitatory neurons. Inhibitory neurons have a more stereotyped relationship between voltage before and after a spike, shorter spike cut lengths, and spike more. The spike cut length and the number of spikes significantly predict a model's capacity to reproduce neuronal spike

times whereas the error in fitting the pre/post-spike voltage did not.

Our work uses similar methods to ref. [20] Both studies incorporate generalizations on the basic LIF model. In addition, both studies implement a fast fitting step followed by a slower optimization of a variable. However, there are some differences: (1) Here, more complex reset rules are used that map the state of the neuron before the spike to that after. (2) We include a membrane potential-dependent adaptation of threshold. (3) We use exponential basis functions for the time-dependencies of the after-spike currents. These allow a much sparser representation, essential for clustering, compared to the non parameterized after spike-current trace composed of wavelets implemented in Pozzorini et al.[20] (4) We use a direct measurement of noise in the membrane potential of a neuron rather than a probabilistic threshold. The exact comparison of performance for models across different publications is difficult because performance is dependent on the properties of the input currents, i.e., if there are large variations in the input current, the capacity to predict the spiking output is better. We chose to compute the performance metrics on stimuli similar to what would be observed in vivo: our stimuli have a coefficient of variation equal to 0.2 centered around baselines at 0.75, 1.0, and 1.25 current threshold (rheobase). The Pozzorini et al.[20] stimulus has a variable coefficient of variation centered around a single baseline. As opposed to comparing models across previous studies, the focus of this study is to apply generalizations of the leaky integrate and fire model to a large database of neuronal responses, characterize the models associated with a large number of transgenic lines, and demonstrate how well they describe the spiking responses and dissociate between cell types.

In an attempt to characterize cell types associated with the input/output transform measured by electrophysiological experiments, clustering algorithms can be run on a set of electrophysiology features. However, it is not entirely clear which are the most important features to consider. An alternative method is to use the entire spike-train by synthesizing the input/output relationship into a model and then performing the clustering on the model parameters. A seemingly intuitive model to use for such a clustering would be a biophysically detailed model, as the

parameters map well to biological mechanisms. However, one problem with such models is that they do not provide unique solutions: repeated optimization with the same input can lead to solutions which are far away in parameter space. Thus, another approach is to use simpler linear models which have both (a) unique parameter solutions, and (b) have been shown to

reproduce spike times of biological data[20]. Indeed, previous studies have touched upon the potential for clustering using simplified models[19].

The unsupervised clusters of the GLIF parameters or the features extracted from the electrophysiological traces do not perfectly match the transgenic lines. This is not surprising given that

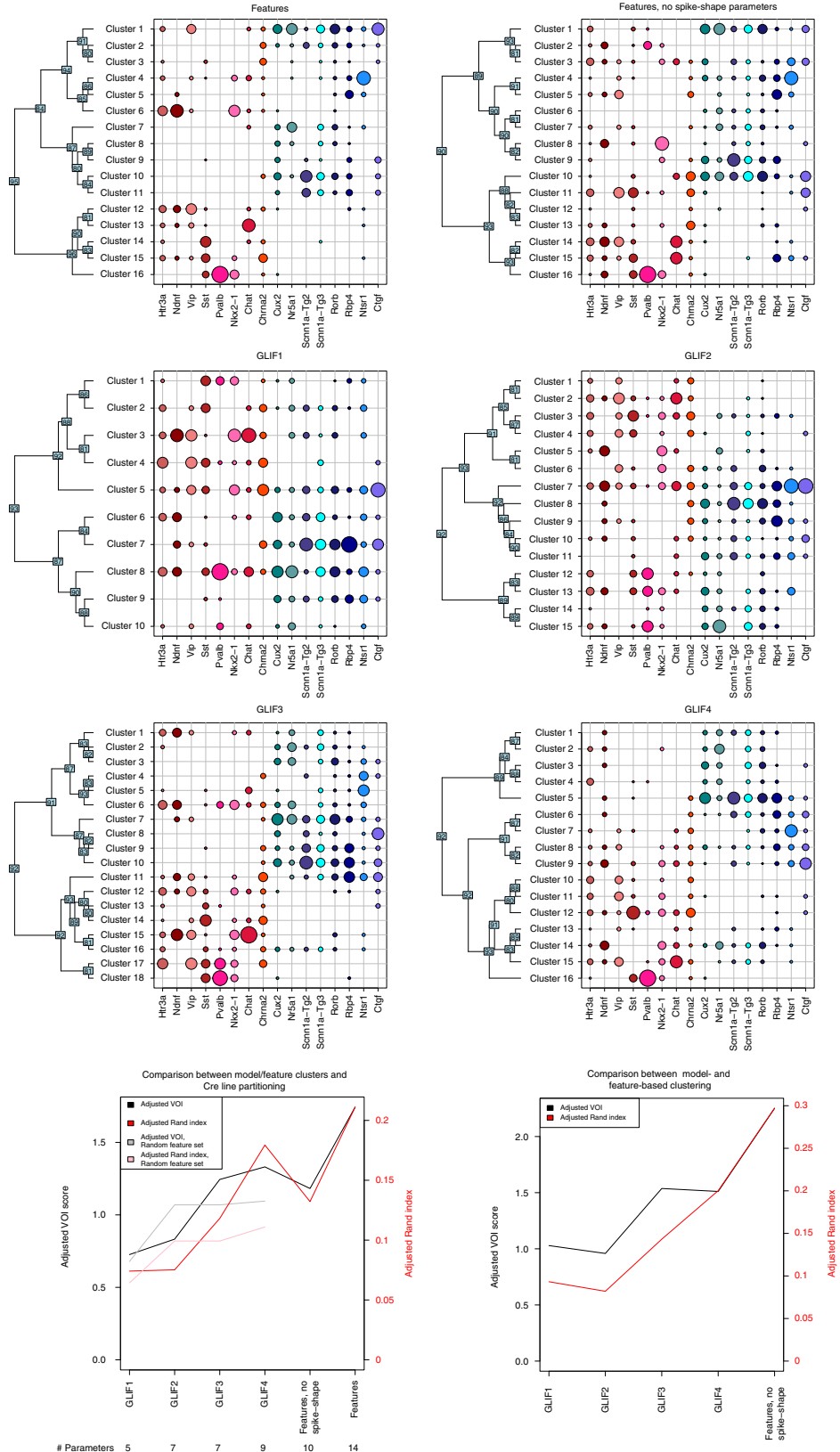

the transgenic lines are known to comprise multiple molecularly defined cell types. In addition, it is not known what specific electrophysiological features are needed to classify cell types. Nonetheless, we certainly expected and did observe relationships.

GLIF model parameters have a better capacity for differentiating transgenic lines than sub-threshold features designed for that purpose. The set of 14 electrophysiological features that include features dependent on spike shape, perform similarly to the $GLIF_4$ parameters we use here, at differentiating transgenic lines (the full set of features perform slightly worse than $GLIF_4$ when differentiated by the iterative binary clustering and perform slightly better when differentiated by affinity propagation). When the spike-shape features are removed, their ability to classify transgenic lines drops below the ability of $GLIF_4$. This suggests that aspects of spike shape are helpful for transgenic line classification[30] when clustering electrophysiological features alone. When including spike-shape features along with $GLIF_3$ and $GLIF_4$ parameters in classification, any improvement in the ability to differentiate Cre lines is slight. This suggests that the $GLIF_3$ and $GLIF_4$ parameters carry much of the information present in this set of sub-threshold and supra-threshold electrophysiological features.

We did not cluster on all available GLIF parameters as they were not available for all 645 cells. It is likely that incorporating the threshold adaptation parameters would further improve clustering. The GLIF model paradigm is sufficiently general that additional mechanisms could be added to the GLIF model family. For example, a mechanism that accounts for the sharp voltage rise at spike initiation[23] incorporated by the adaptive exponential integrate and fire model[31] could further raise the clustering capabilities of GLIF parameters.

As with all abstractions, there are limitations to our models: the traces which are reproduced are coming from a current injection into the soma of the neuron under stereotyped recording conditions. We try to keep the statistics of the test stimulus similar to in vivo patch clamp currents, but in vivo, additional large sources of variability will occur from stochastic transmission at synapses, complex dendritic integration, neuromodulation, etc. These mechanisms, which are necessary for exact integration of single-neuron models into systems models are beyond the scope of the current study.

For any study, the best model to use will depend on the question at hand. Currently, it is unknown what level of spike precision will be needed in network models. However, as the brain has evolved a diversity of cell types, it is likely cell type-specific models will be necessary to model brain computation. We hope that this study along with our corresponding publicly available database of models will be a useful resource for the community by providing both optimized models and an intuition

for how much complexity aids in spike time performance and cell type classification.

## Methods

**Model definitions.** We developed five different GLIF models, each of which successively incorporates the time evolution of one to five state variables. For brevity, here all model level equations are presented together. A separate and detailed description of the models along with their mathematical equations are available in the "Model definitions" section of the Supplementary Methods.

The set of five state variables $X = \{V(t), \Theta_s(t), I_{(j=1,2)}(t), \Theta_V(t)\}$ represent the neuronal membrane potential, the spike-dependent component of the threshold, two after-spike currents and the membrane potential-dependent component of the threshold, respectively. Figure 1a illustrates the iteration between the state variables in the model equations. Table 1 contains a summary of the parameters and variables. It is assumed that these state variables evolve in a linear manner between spikes:

$$V'(t) = \frac{1}{C}\left(I_e(t) + \sum_j I_j(t) - \frac{1}{R}(V(t) - E_L)\right), \quad (1)$$

$$\Theta_s'(t) = -b_s\Theta_s(t), \quad (2)$$

$$I_j'(t) = -k_j I_j(t); \; j = 1, 2, \quad (3)$$

$$\Theta_v'(t) = a_v(V(t) - E_L) - b_v\Theta_v(t), \quad (4)$$

where $C$ represents the neurons's capacitance, $R$, the membrane resistance, $E_L$, the resting membrane potential, $I_e$, the external current, $1/k_j$, $1/b_s$, $1/b_v$ are the time constants of the after-spike currents, spike and voltage dependence of the threshold, and $a_v$ couples the membrane potential to the threshold.

If $V(t) > \Theta_v(t) + \Theta_s(t) + \Theta_\infty$, a spike is generated. After a refractory period, $\delta_t$, the state variables are updated with a linear dependence on the state before the spike:

$$V(t_+) \leftarrow E_L + f_v \times (V(t_-) - E_L) - \delta V, \quad (5)$$

$$\Theta_s(t_+) \leftarrow \Theta_s(t_-) + \delta\Theta_s, \quad (6)$$

$$I_j(t_+) \leftarrow f_j \times I_j(t_-) + \delta I_j, \quad (7)$$

$$\Theta_v(t_+) \leftarrow \Theta_v(t_-). \quad (8)$$

Where $t_+$ and $t_-$ represent the time just after and before a spike, respectively. $f_v$ and $\delta_V$, are the slope and intercept of the linear relationship of the voltage before and after a spike (see Supplementary Fig. 1). $\delta\Theta_s$ is the amplitude of the spike-induced threshold after spiking. $f_j$, is a fraction of current implemented after a spike; here it is always set to 1. $\delta I_j$, is the amplitude of the spike-induced currents.

The 1D model, $GLIF_1$, is represented by equation 1, with a simple reset and $I_j(t) = 0$ for all j. $GLIF_2$ adds $\Theta_s$ and the more complex reset rules (Eqs. 5 and 6). $GLIF_3$ uses membrane potential and two spike-induced currents, $I_j(t)$ in Eq. 1, and reset rules defined by Eq. 7. $GLIF_4$ uses membrane potential, spike-induced threshold dependence and after spike currents, whereas $GLIF_5$ uses all the variables described.

**Fig. 6** We identify discrete putative clusters using an iterative binary clustering approach on 645 cells. The top six panels show the summary of clusters obtained by iterative binary clustering using electrophysiological features extracted from the traces and GLIF model parameters. In every panel, each row represents a cluster, and each column a transgenic line. The size of the circle indicates the fraction of cells from a given transgenic line falling into a specific cluster (such that the sum of fractions in a column add up to 1). The dendrogram on the y-axis shows the iterative binary splitting into clusters using the algorithm explained in the text. For each intermediate node, a support vector machine was trained on half the cells at that node and used to classify the remaining cells. The number at each node indicates the minimum percentage of test cells correctly classified over 100 iterations of randomly selected training and test cells. Clustering based on features and using the $GLIF_3$ and $GLIF_4$ model parameters shows separation among lines labeling inhibitory and excitatory cells. In addition, transgenic Cre lines marking Pvalb+, Ntsr1+, Nr5a1+, and Ctgf+cells tend to segregate into distinct clusters. The bottom two panels show two measures of overall clustering similarity: the adjusted Rand index (ARI) in red, and the adjusted variation of information (AVOI) in black. The bottom left panel shows similarity between each set of clusters and the transgenic lines. The bottom right panel shows similarity between each set of clusters and the clusters obtained using the features. An ARI of 1 indicates perfect agreement between partitions, whereas 0 or negative values indicate chance levels of agreement. A positive value of the AVOI indicates agreement between partitions that is better than chance (which is indicated by 0). The gray and pink traces in these two panels show the AVOI and ARI values, respectively, for random subsets of the features containing the same number of parameters as each of the four GLIF models

**Stimulus**. We designed a set of four different types of stimuli for fitting and testing parameters of our GLIF models (Fig. 2). This set of stimuli is part of a much larger electrophysiological stimulation protocol which are the subject of different studies. Details of the stimuli described here and the other stimuli of the Allen Cell Type Database can be found at ref. [24]

The first type of stimulus we used were very short (3 ms) square pulses of multiple amplitudes to assess the threshold of a neuron. The minimal voltage at which the neuron fires a spike in response to a short current pulse is defined as the instantaneous threshold, $\Theta_\infty$. This parameter is later optimized in the post hoc procedure (see sections, "Fitting and post hoc optimization of instantaneous

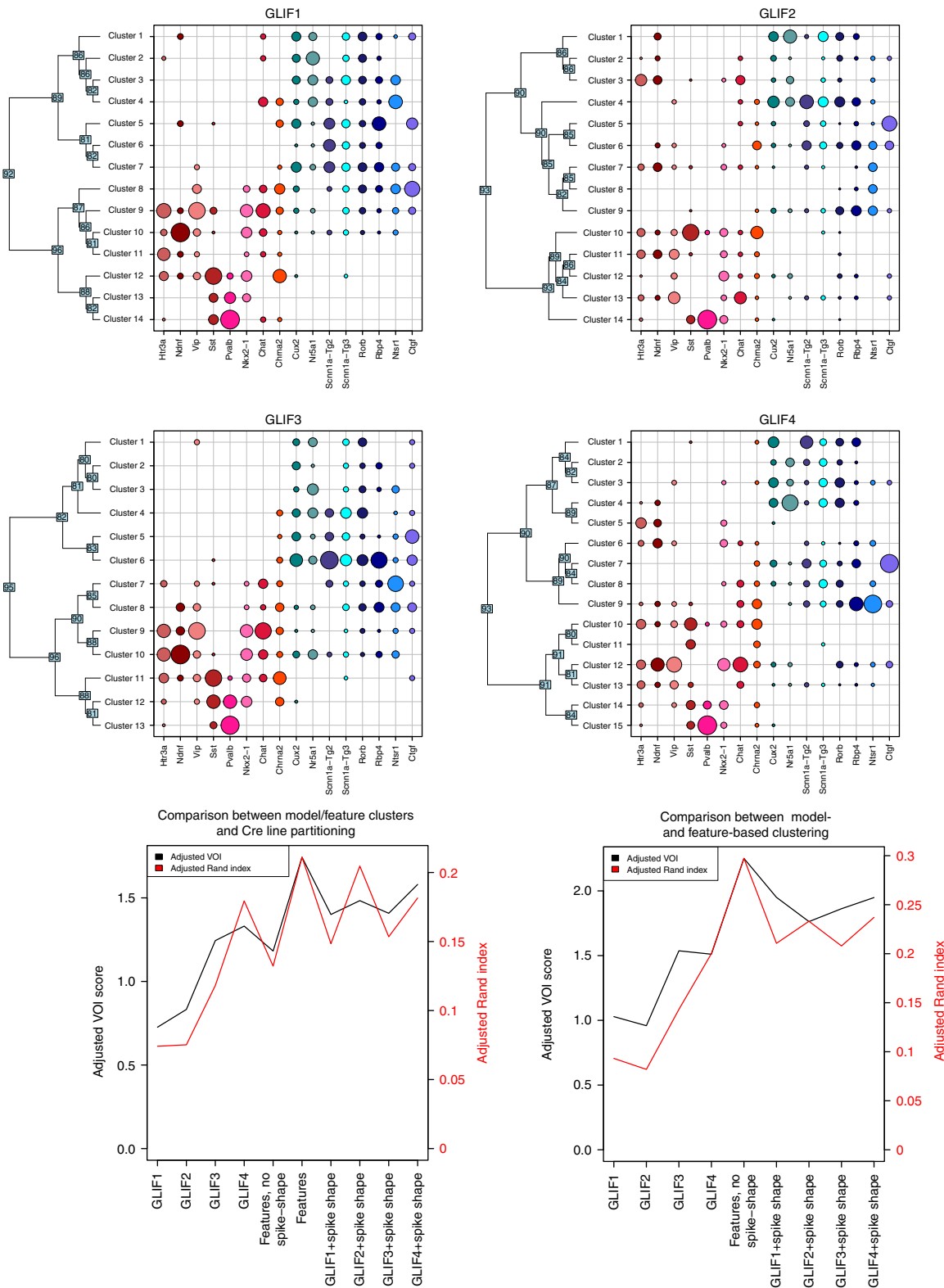

**Fig. 7** Iterative binary splitting clustering obtained from using GLIF model parameters plus spike-shape-related feature parameters. The top four panels show cluster versus Cre line composition, similar to Fig. 6. The bottom two panels show the adjusted variation of information metric and the adjusted Rand index for the GLIF model-derived clusters with and without the spike-shape parameters

threshold in the main text", and "Post hoc optimization" in the Supplementary Methods).

The most substantial type of stimulus we used, consists of noise stimuli. This noise was created to have a coefficient of variation of 0.2 (similar to in vivo patch clamp recordings) at three different base amplitudes: one sub-threshold, one peri-threshold and one supra-threshold to explore different parameter regimes. Each amplitude is given for 3 s with a 5 s break in between to allow the neuron to settle to a rest state. Using stimuli with highly varied structure is important. Not only does it allow the model to explore the parameter space but it is more realistic to what a neuron actually experiences in vivo. Solely using traditional simple stimuli, such as square pulses, would present only a small subset of possible histories (both spiking and sub-threshold) and bias the data set to a non-physiological regime. Within this noise data we have two different complete sets of noise which we refer to as noise 1 and noise 2. The noise 1 and noise 2 stimuli have the same statistics but were created with a different random seed. We used noise 1 data to fit the model and noise 2 as "hold out" test data to ensure we have not over fit our model. Both the noise 1 and noise 2 stimuli are repeated at least twice to characterize how consistently the biological neuron fires at the same time for a repeated stimulus.

The third type of stimulus is a (3 s) sub-threshold long square pulse just below rheobase given to characterize the intrinsic noise in the neuronal membrane near threshold (please see "Objective function: maximum likelihood based on internal noise (MLIN)" in the Supplementary Methods and Supplementary Fig. 5).

Finally, a series of short square pulses at different frequencies, referred to as a "triple short square" (Supplementary Fig. 3), are used to characterize the spike-induced changes in threshold in the absence of additional membrane potential-induced changes (such as the voltage-induced changes in threshold modeled in $GLIF_5$). This stimulus is necessary to fit all models with a spike component of threshold, i.e., $GLIF_2$, $GLIF_4$, and $GLIF_5$. Due to experimental constraints independent of this study, this triple short square stimulus was only played to approximately half the neurons. This is why there are many fewer $GLIF_2$, $GLIF_4$, and $GLIF_5$ models available.

**Model exclusion criteria**. Only transgenic lines which contained more than five neurons and had the necessary data to create a basic $GLIF_1$ model described in "Stimulus" section, were included in this study. In addition, models which had biologically unrealistic parameter values after fitting, or neurons that had an explained variance ratio value less than 20% for one model on the training data set (noise 1) were eliminated (please see the "Exclusion criteria" section in the Supplementary Methods for a detailed explanation of excluded data). Overall, a total of 645 neurons from 16 transgenic lines met the criteria for fitting the $GLIF_1$ and $GLIF_3$ models. As mentioned in the "Stimulus" section, only some neurons were given the stimulus needed to create $GLIF_2$, $GLIF_4$, and $GLIF_5$ models. 254 neurons met the additional requirements for models with reset rules ($GLIF_2$ and $GLIF_4$) and 253 neurons met all the criteria for a $GIF_5$ model.

**Fitting and post hoc optimization of instantaneous threshold, $\Theta_\infty$**. Similar to ref.[20], GLIF model fitting is achieved in two steps. In the first step, parameters are fit directly from the electrophysiological data using linear methods described in detail in the "Parameter fitting and distributions" section of the Supplementary Methods: $E_L$ from the mean membrane potential at rest, $R$, and $C$ from fitting the membrane potential during a sub-threshold noise stimulus, $\delta I_j, k_j$ from fitting the membrane potential between spikes during a supra-threshold stimulus, $f_v, \delta t, \delta V$ relating the potential before and after a spike during the noise stimulus, $b_s, \delta \Theta_S$ from the potential of subsequent spikes during a series of brief square pulses, $a_v$ and $b_v$ from the potential of subsequent spikes during the supra-threshold noise stimulus. Slices from this parameter space are shown in Fig. 3a–f.

In the second step, the instantaneous threshold, $\Theta_\infty$, was optimized to fit the probability of a neuron model reproducing observed spike times. This is realized using a non-linear Nelder Mead optimization strategy as described in detail in the "Post hoc optimization" section of the Supplementary Methods. $\Theta_\infty$ represents the threshold of a neuron when it is stimulated from rest. Any parameter, or combination of parameters, could have been chosen for additional optimization. However, the time needed to optimize more than one parameter is prohibitive. $\Theta_\infty$ was chosen for further optimization because it is represents the overall excitability of a neuron and therefore is likely to be the best parameter to counteract any error in the fitting or the inherent error introduced by simplifying any complex system to a simple model. The changes in instantaneous threshold, $\theta_\infty$, with optimization subsection of the Supplementary Methods and Supplementary Fig. 7 discuss how $\Theta_\infty$ is altered by optimization.

**Electrophysiological features**. We use a set of 14 electrophysiological features to perform clustering. Detailed explanations of these features can be found at ref.[24]. Here, we briefly describe them below.

Membrane time constant ($\tau_m$): The membrane time constant of the cell was calculated by averaging the time constants of single-exponential fits to hyperpolarizing responses evoked by one-second current steps with amplitudes from −10 pA to −90 pA (20 pA interval). Fits were performed on the

traces from 10% of the maximum voltage deflection to the maximum voltage deflection.

Input resistance ($R_i$): The input resistance was calculated by first measuring the peak voltage deflection evoked by one-second current steps with amplitudes from −10 pA to −90 pA (20 pA interval), then taking the slope of a linear fit to those deflections versus the respective stimulus amplitudes.

Resting potential ($V_{rest}$): The resting potential was calculated by averaging the pre-stimulus membrane potential across all one-second current step responses.

Threshold current ($I_{thresh}$): The threshold current was the minimum amplitude of a one-second current step that evoked at least one action potential.

Action potential threshold ($V_{thresh}$): The action potential threshold was defined as the membrane potential before the action potential peak where the d$V$/d$t$ was 5% of the maximum d$V$/d$t$ averaged across all spikes evoked by the stimulus.

Action potential peak ($V_{peak}$): The action potential peak was defined as the maximum membrane potential reached during an action potential.

Action potential fast trough ($V_{fasttrough}$): The action potential fast trough was defined as the minimum value of the membrane potential within 5 ms after the action potential peak.

Action potential trough ($V_{trough}$): The action potential fast trough was defined as the minimum value of the membrane potential before the next spike (or end of the stimulus interval).

Upstroke/downstroke ratio (up:downstroke): The ratio between the absolutes values of the action potential peak upstroke (i.e., maximum d$V$/d$t$ value before the peak) and action potential peak downstroke (i.e., minimum d$V$/d$t$ value after the peak). Here, the upstroke/downstroke ratio was separately measured on action potentials evoked by a one-second (long square) current step and a 3 ms (short square) current step.

Sag: The sag was calculated as the difference between the minimum membrane potential value reached during a hyperpolarizing one-second current step and the steady-state membrane potential during that step, divided by the difference between the minimum membrane potential value reached during the step and the baseline membrane potential. The sag was calculated on the step where the minimum membrane potential was closest to a value of −100 mV.

$f$–$I$ curve slope: The $f$–$I$ curve slope was calculated by measuring the average firing rate during one-second current steps versus the stimulus amplitude. The supra-threshold part of the curve was fit with a line, and the slope was taken from that linear fit.

Latency to the first action potential (latency): Latency was calculated as the interval between the start of the stimulus and the time of the first spike evoked by the stimulus.

Maximum burst index (max. burst index): If a burst was identified during a response, a burst index was calculated as the difference between the maximum instantaneous firing rate inside the burst and the maximum instantaneous firing rate outside the burst, normalized by their sum. Bursts were identified as a change in the character of the voltage trajectory during the interspike interval (e.g., changing from a "direct" trajectory, where the membrane potential was always increasing after a spike, to a "delay" trajectory, where the membrane potential first hyperpolarized, then depolarized after a spike). If no burst was detected, the index was zero. The maximum index reported here was the maximum across all the supra-threshold responses evoked by depolarizing one-second current steps.

**Data availability**. All data are publicly and freely available on the Allen Cell Type Database at http://celltypes.brain-map.org. Analysis code is available in the Allen Institute Github repository at https://github.com/AllenInstitute/GLIF_Teeter_et_al_2018. Code for the creation of the pink noise stimulus can be found at https://github.com/AllenInstitute/ephys_pink_noise.

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

## Acknowledgements

The authors thank the Allen Institute founder, Paul G. Allen, for his vision, encouragement, and support. We thank the following people for their contributions to the work enabling our project: Lydia Ng for technology team leadership, Amy Bernard and John Phillips for data production team leadership, and Susan Sunkin for program and project management support. We thank all those on the data production and technology teams who helped to collect, curate, quality control and make the data publicly available.

## Author contributions

C.T. developed the project, created and implemented the fitting, optimization, pipeline and analysis codes, developed the algorithms, preformed the mathematical derivations, developed and programmed the experimental stimuli, and wrote, edited and is responsible for all subjects of the manuscript. R.I. performed mathematical derivations, developed and created the code for the generalized linear model (GLM) used in fitting, developed the experimental pink noise stimuli, and wrote and edited the GLM subject. V.M. is responsible for the intellectual and algorithm development as well as code implementation, writing, and editing of the clustering analyses. N.G. is responsible for the conceptualization, algorithm development, code implementation, and writing and editing of the electrophysiological feature extraction analyses. D.F. developed, implemented, and deployed the model pipeline. J.B. was the manager of electrophysiological pipeline data collection and wrote and edited the electrophysicological and cre-line subjects. A.S. was responsible for data curation and public release. N.C. time-optimized the post hoc optimization algorithm. H.Z. conceived and managed electrophysiological data production pipeline. M.H. directed the primary group conducting this research. C.K. initiated the project, provided overall scientific guidance and edited all subjects of the manuscript. S.M. conceived and developed the modeling project, created the models, performed mathematical derivations, provided scientific guidance, conceptualized and developed the experimental stimuli, and wrote, edited, and is responsible for all subjects of the manuscript.

## Additional information

**Competing interests:** The authors declare no competing financial interests.

