## [Peer Review File · Nature Communications]

Reviewers' comments:

Reviewer #1 (Remarks to the Author):

The authors describe several classes of generalized leaky integrate-and-fire models of increasing complexity, and fit these models to the Allen Institute cell types dataset (celltypes.brain-map.org/) with the aim of reducing the dimensionality of these cells into a few biologically interpretable parameters. They subsequently use these fit parameters to cluster the data and compare the resulting clusters to alternative clusters based on electrophysiological features.

There is interesting information in this study (extensions of generalized integrate and fire models, applications to the Allen institute data), but it was hard for me to discern the main fundamental contribution. In general this contribution should be described more directly. Specifically, the aim of this paper does not seem the description of the Allen cell-type data per se, while the use of generalized leaky integrate and fire to reduce dimensionality has already been described in Pozzorini (2015). In this context, it seems that the major (i.e. non-incremental) novelty here is the clustering of the data based on the model parameters. But in this case it would benefit if the authors expanded on the description and validation of this aspect of the study in much greater detail. I discuss this and other comments below.

It seems that the main advantage of the present framework is that the characteristic parameters of cells in each cluster may intuitively describe the activity of cells in the cluster. However, the parameter space is quite large and so it becomes hard to eyeball the characteristic property of each cluster, based on these parameters at a glance. The authors seem to justify the use of such a large number of parameters based on the number of clusters produced by the algorithm, arguing that a large number of parameters is required to have a larger number of clusters. But, these considerations are based on one specific clustering algorithm and are not so convincing. The output of the number of clusters for an algorithm is a difficult problem without a clear solution: here, the authors base it on the likelihood associated with the data being derived from two multivariate gaussians versus a single multivariate gaussian, without actually providing evidence that the assumption of gaussianity is warranted (and an eyeball of some of the parameter distributions provided by the authors shows strong evidence of non-gaussianity).

More generally, there are many clustering algorithms based on independent assumptions for the number of clusters and so it is difficult to conclude that the number of clusters based on the authors' present analysis is definitive. Arguably the objective here should be to find the clustering closest to a clustering based on electrophysiological features, rather than a clustering with the largest number of clusters. It does not seem that a distance between clusterings was computed, but this is crucial if the authors wish to show that their clustering is indeed similar to other types of dimensionality reduction. Statements such as "among the excitatory-neuron-dominated clusters, there is more similarity among the electrophysiological features, but some relevant differences can be observed." are not in themselves sufficient.

In this context, here are some recommendations:

- 1) I suggest that the authors provide more evidence for the size and composition of their clusters, using complementary algorithms if necessary. Just to give an example from the recent literature, affinity propagation has been used to successfully cluster neurons based on morphology (Costa et al. *Neuron*. 2016 Jul 20; 91(2):293-311. doi: 10.1016/j.neuron.2016.06.012) and could also be relevant for the present data.
- 2) The number of clusters is not so interesting per se, cf. the similarity of the clusters to clusters based on electrophysiological features. It would be interesting to see whether if clustering based on

more complicated models reveals a consistent increase in such similarity.

3) Given the above, can the authors find a smaller subset of parameters, which while not necessarily explaining as much variance of individual neurons, provides the most reliable clustering of the models into underlying neuronal subtypes? The advantage of a smaller subset parameters is increased ease in interpretability and robustness.

Other comments:

* What is the motivation for the specific generalizations of the models chosen by the authors. For instance, Izhikevich (2004) provided a comprehensive list of interesting neuronal properties. The authors have chosen to incorporate some of these properties, but the specific choice has not really been motivated. In addition, not all possible choices of model combinations have been explored (it is possible to consider other types of models such as GLIF2 which couple membrane potential to the threshold but include none of the other model generalizations). The present generalizations are thus presented and motivated in a somewhat ad hoc fashion. Furthermore, it seems that no formal model selection was performed (the variance is bound to increase with the complexity of the model, but the complexity vs fit tradeoff has not been formally quantified).

* The description of Θ_{inf} is confusing, despite the fact that this parameter seems to be one of the most important to the observed fit. For example, the formal definitions of the models in section 2.1 do not actually include Θ_{inf} anywhere, but at the same time the authors include Θ_{inf} into the threshold criterion without describing what this parameter exactly represents. If this parameter is indeed a constant, what is its biological interpretation? Could another potentially ad hoc parameter have been used instead of Θ_{inf} to improve the fit? The reason for post-hoc optimization of this parameter should be in the main text rather than SI. Furthermore the importance of the statement "although Θ_{inf} did change between the fitting and posthoc optimization step, on average the value remains consistent" is unclear because the individual values clearly changed given the improved fits.

* The authors actually remove quite a lot of neurons not suitable for fitting to their more complex models (a reduction from >700 for GLIF1 to <300 for some of the more complex models). How useful can these models be if only a minority of the neurons can be meaningfully fit?

* Could the authors comment on why the inhibitory cf. excitatory neurons were fit consistently better?

* How much are the more complex models hampered by lack of fit? For instance, GLIF2 performance is actually reduced cf. GLIF1, presumably because the parameter fit is worse. What is the main determinant of the poor fit in this case, Θ_{inf} ? This should be discussed in more detail.

* Figure 5. It would make more sense to group each type of neuron separately (excitatory, inhibitory, and all neurons). The effect sizes of the models within this groups are more important than the p values and these effect sizes will be easier to ascertain through such a grouping.

* The presentation in general is very sloppy. For example, I picked up the following text mistakes during my review, and there could be others: eletrophysiological, neurons's , during during, Supplimentary, synthesizing, distinct, and and an, Columns 1 though 5, cells and and neurons, can be rearranges as follows, Can an be seen. (for Brain Science, 2016) is used as the last name of an author, capital and lower theta are used interchangeably in equations, etc. I urge authors to thoroughly clean up the presentation of the manuscript.

Reviewer #2 (Remarks to the Author):

The authors fit a neuron model with varying complexity to a large database of electrophysiological recordings and assess whether adding complexity to the model improves its ability to reproduce spike trains in response to noisy input and attempt to cluster the neurons into classes that overlap with the different cell lines based on the parameters of the model. They conclude that added complexity improves performance and model parameters provide a better classification compared to features extracted directly from voltage traces.

While I appreciate the approach to assess the utility of a neuron model with the impressive database of the Allen Institute, I do not see that this study constitutes a substantial addition to the literature. My main concern is on methodological issues. The two main results are not supported in a statistical proper way. The improvements of the model performance is assessed using a number of pair comparisons with no apparent correction for alpha error inflation (a Kruskal-Wallis test would be appropriate here). From visual inspection of the boxplots in Figures 5, it is clear that the different models do not notably differ in performance, so even if there are any statistically significant differences, they are of no practical significance. Similarly, the improvement in clustering the cell lines using the model parameters is claimed to be better compared to clustering based on the voltage trace features, but no quantitative support of this claim is given beyond a superficial examination of Figure 6.

A second concern is on the significance and novelty of the results. Comparisons of model performance have been performed before (e.g. Jolivet et al. 2008) and the same is true for clustering approaches to neuron classification (e.g. Ardid et al. 2015), so I don't see the exact contribution of this manuscript.

A possible way the authors could go may be a comparison of established neuron models (e.g. in comparison with the GLIF variants) in a statistically sound way (and using more appropriate ways of comparing them, e.g. Bayesian model selection, which incorporates penalties for model complexity). Their rich database of electrophysiological recordings certainly provides a unique resource for such a useful endeavor.

Reviewer #3 (Remarks to the Author):

The models and approach presented in this paper are a valuable resource. The study is conceptually not that different from prior GLIF model development studies, but its use of a very extensive dataset sets it apart in implementation. The major result is that GLIF models perform well, and that their parameter space clusters similarly to that of the physiological data.

Major comments.

1. I feel this is more of a resource study than a research article. As a resource, this is in line with the other valuable resources of the Allen Institute. It has brought together a vast amount of data with careful model fitting and testing in a range of GLIF detail.

I'm not sure there is yet a strong case that having biologically accurate firing patterns in integrate-and-fire models makes a huge difference to network computation. Nevertheless, I expect that the reported models will find wide use. The analysis on how much is gained by different levels of GLIF model will also be useful.

2. I don't gain much physiological or computational insight from this study. There is the intriguing point that these models converge during parameter fits, in contrast to the observation that detailed ones have multiple solutions (Prinz and Marder). But I feel that is mostly a reflection of these models rather effectively mapping physiological traces to parameters.

Minor points:

Figure 1 conveys rather complicated and extensive information, and the figure is overwhelmed by its very long legend. It isn't even referenced in the text. Possibly the authors may consider having a regular introductory section in the main text that refers to the figure?

I was looking to see the experimentally observed spiking pattern in panel (b). It is all very well for the GLIFs to converge to a certain spiking pattern, but how close is this to the real thing? This comparison shows up later in Figure 4,9.

Page 3, 4 lines from bottom: typo: "eletrophysiological"

Figure 6 legend: "all optimized model parameters 8"

Figure 6: There is no panel d, though it is referred to in the legend.

Figure 11 legend "Can an be seen"

Dear Reviewers,

We thank you for taking the time to provide valuable feedback on our manuscript. A summary of the changes we have made in the revised manuscript are below:

- A) We have rewritten the text for clarity, readability, and to closer adhere to the standards of Nature Communications.
- B) We provide clarification on:
 - I) our contribution to the field.
 - II) why we chose our specific models.
- C) We have updated our statistic including:
 - I) We correct for alpha error inflation using the Benjamini-Hochberg correction.
 - II) We have included Mann Whitney U tests on the distribution of differences between different GLIF levels to show that the improvement or decrement between GLIF levels is different for excitatory and inhibitory neurons.
- D) We have removed the gaussian mixture model from our hierarchical clustering method (thus removing any issue with non-gaussian parameter distributions).
- E) We have included an additional clustering method (affinity propagation) to substantiate our clustering results.
- F) We have included a similarity metric (the adjusted rand index) to quantify the similarities between GLIF clusters, feature clusters and transgenic lines.
- G) We have updated our data set.

In our revised manuscript we are clearer about our exact contribution and novel discoveries. Details of these contributions and how they fit into the field are available in the main text. Briefly, we show:

- A) GLIF clusters can differentiate transgenic lines better than subthreshold electrophysiological features. This is the first work that has shown the ability of any generalization on the leaky integrate and fire model to classify different cell types associated with transgenic lines.
- B) Different phenomenological mechanisms are needed to recreate the firing patterns of excitatory and inhibitory neurons.
- C) The ability of GLIF parameters to differentiate between different transgenic lines follows the same trend as the ability of neurons to reproduce spiking behavior. Thus we suspect optimizing simple models to recreate spiking behavior identifies relevant parameters for cell type identification.
- D) Quantify the ability of different phenomenological generalizations fit from the data to both recreate spike times and differentiate transgenic lines. This provides an idea of the scale of improvement generalizations on the traditional LIF model yield and will be helpful to those creating network models, often which incorporate the use of traditional LIF models.
- E) Show that inhibitory neurons are easier to fit than excitatory neurons and provided evidence that this is because inhibitory neurons are more stereotypical.

F) Although it may seem intuitive that increasing model complexity by including mechanisms fit to aspects of the voltage waveform would improve spike time reproducibility, this is not always the case.

Please note that our data set has changed in this version of the manuscript. This is because the Allen Institute has extensive quality control (QC) criteria neurons must pass to be made publicly available. Many of the neurons we used in the first version of the manuscript were not yet publicly released and ended up failing a QC step that takes place after electrophysiology (for example at the imaging step). Also, additional neurons have been recorded which have passed all QC criteria. All data in this manuscript is either already available or will be available in the Allen Institute October data release. All analysis code will also be made available.

We have answered your specific questions inline below. We greatly appreciate your time and consideration and look forward to your assessment of our revised manuscript.

Sincerely,
Teeter, et al.

Reviewers' comments:

Reviewer #1 (Remarks to the Author):

The authors describe several classes of generalized leaky integrate-and-fire models of increasing complexity, and fit these models to the Allen Institute cell types dataset (celltypes.brain-map.org/) with the aim of reducing the dimensionality of these cells into a few biologically interpretable parameters. They subsequently use these fit parameters to cluster the data and compare the resulting clusters to alternative clusters based on electrophysiological features.

There is interesting information in this study (extensions of generalized integrate and fire models, applications to the Allen institute data), but it was hard for me to discern the main fundamental contribution. In general this contribution should be described more directly. Specifically, the aim of this paper does not seem the description of the Allen cell-type data per se, while the use of generalized leaky integrate and fire to reduce dimensionality has already been described in Pozzorini (2015). In this context, it seems that the major (i.e. non-incremental) novelty here is the clustering of the data based on the model parameters. But in this case it would benefit if the authors expanded on the description and validation of this aspect of the study in much greater detail. I discuss this and other comments below.

The aim of this manuscript is to both describe the GLIF portion of the Allen Cell Type Database and demonstrate the scientific discoveries we have made with the models. The Pozzorini et al. (2015) paper showed that fitting GIF models could be automated to fit biological data. Here we show that GLIF parameters can differentiate transgenic lines using unsupervised learning. To our knowledge, this is a novel contribution to the field.

Mensi et al. (2012) showed the potential of GLIF parameters to differentiate cell types. They showed the GLIF parameters of three, layer 5 neuron types (GABAergic fast spiking, GABAergic regular spiking, and excitatory) could be differentiated using PCA and a linear classifier. They were definitely on the right track: here we show the potential of unsupervised learning on GLIF parameters to differentiate transgenic lines on hundreds of cells. We show that a limited set of GLIF parameters can differentiate transgenic lines better than subthreshold electrophysiological features.

It seems that the main advantage of the present framework is that the characteristic parameters of cells in each cluster may intuitively describe the activity of cells in the cluster. However, the parameter space is quite large and so it becomes hard to eyeball the characteristic property of each cluster, based on these parameters at a glance.

We believe that one of the major contributions of our manuscript is that we show GLIF parameters can differentiate transgenic lines better than subthreshold electrophysiological features. This has not been previously shown.

The authors seem to justify the use of such a large number of parameters based on the number of clusters produced by the algorithm, arguing that a large number of parameters is required to have a larger number of clusters. But, these considerations are based on one specific clustering algorithm and are not so convincing. The output of the number of clusters for an algorithm is a difficult problem without a clear solution: here, the authors base it on the likelihood associated with the data being derived from two multivariate gaussians versus a single multivariate gaussian, without actually providing evidence that the assumption of gaussianity is warranted (and an eyeball of some of the parameter distributions provided by the authors shows strong evidence of non-gaussianity).

This is an excellent point – we originally used a combination of a multivariate Gaussian likelihood method (sigClust) and a support vector machine as a check. Given that the distributions of some of the parameters are indeed highly non-Gaussian, we removed the multivariate Gaussian likelihood criterion, and use only the support vector machine as a check to establish the reliability of each binary split. The requirements for the SVM are stringent: we use half the cells for training, and require that the test set (the remaining half) be assigned with >80% accuracy for each of the two classes. Originally, we did use the multivariate gaussian likelihood and then verified it with an support vector machine as a check.

In addition, we include the affinity propagation clustering method to substantiate our claims.

More generally, there are many clustering algorithms based on independent assumptions for the number of clusters and so it is difficult to conclude that the number of clusters based on the authors' present analysis is definitive. Arguably the objective here should be to find the clustering closest to a clustering based on electrophysiological features, rather than a clustering with the largest number of clusters. It does not seem that a distance between clusterings was

computed, but this is crucial if the authors wish to show that their clustering is indeed similar to other types of dimensionality reduction. Statements such as "among the excitatory-neuron-dominated clusters, there is more similarity among the electrophysiological features, but some relevant differences can be observed." are not in themselves sufficient.

It was not our intention to use number of clusters as our metric of good clustering although it did appear that way. We wanted to assess the ability of GLIF parameters to differentiate between transgenic lines. Since there were 14 transgenic lines in the first study (there are 16 in this version), we expected around 14 clusters assuming the different transgenic lines have distinct electrophysiological phenotypes, and even more clusters if the GLIF models could distinguish subclasses of electrophysiological phenotypes within transgenic lines. We were struggling to find a way to describe our clustering results which we have now quantified with the Adjusted Rand Index in this revised manuscript.

Our objective was not necessarily to find the clustering closest to a clustering based on ephys features as there is not a set of ephys features that are 'ground truth' in clustering. However, we would expect a relationship between GLIF parameters clusters and ephys feature clusters. In our previous manuscript draft, we tried to convey that relationship with confusion matrices. In this version, will still provide the confusion matrices, but also quantify this relationship with the Adjusted Rand Index.

In this context, here are some recommendations:

1) I suggest that the authors provide more evidence for the size and composition of their clusters, using complementary algorithms if necessary. Just to give an example from the recent literature, affinity propagation has been used to successfully cluster neurons based on morphology (Costa et al. Neuron. 2016 Jul 20;91(2):293-311. doi: 10.1016/j.neuron.2016.06.012) and could also be relevant for the present data.

We have included the suggested affinity propagation clustering method to substantiate our results.

2) The number of clusters is not so interesting per se, cf. the similarity of the clusters to clusters based on electrophysiological features. It would be interesting to see whether if clustering based on more complicated models reveals a consistent increase in such similarity.

We now include the Adjusted Rand Index as a measure of similarity between different clustering paradigms. As you will see in the revised manuscript, more complexity does not necessarily mean better performance (in clustering or spike time reproduction).

However, the model's ability to differentiate different cre-lines follows the same trend as their ability to reproduce spike trains!

3) Given the above, can the authors find a smaller subset of parameters, which while not necessarily explaining as much variance of individual neurons, provides the most reliable clustering of the models into underlying neuronal subtypes? The advantage of a smaller subset parameters is increased ease in interpretability and robustness.

We did not specifically look for a minimal set of parameters. However, we do quantify the ability of GLIF model parameters to differentiate different cre lines and show that it follows the same trend same trend as their ability to reproduce spike trains.

Other comments:

What is the motivation for the specific generalizations of the models chosen by the authors. For instance, Izhikevich (2004) provided a comprehensive list of interesting neuronal properties. The authors have chosen to incorporate some of these properties, but the specific choice has not really been motivated. In addition, not all possible choices of model combinations have been explored (it is possible to consider other types of models such as GLIF2 which couple membrane potential to the threshold but include none of the other model generalizations). The present generalizations are thus presented and motivated in a somewhat ad hoc fashion.

We can understand how our overall choice of mechanisms and model choice could have been somewhat mysterious in our original submission. We have edited the text to make it clearer. We chose to limit our investigation to generalizations of the LIF model which can be expressed as linear dependencies in the reset rules (R), or linear dependencies in the dynamics which are membrane potential independent (after-spike currents) or membrane potential-dependent (voltage-dependent threshold).

Furthermore, it seems that no formal model selection was performed (the variance is bound to increase with the complexity of the model, but the complexity vs fit tradeoff has not been formally quantified).

For model selection we used the 'gold standard' in the field: the capacity of the model to predict responses on a "hold out" testing dataset, which was not used at all in the model construction. It should be noted that a more complex model does not necessarily result in improved spike time performance.

We have added an additional calculation of the Akaike Information Criterion (AIC) calculated on the training data set (which would usually be done in situations where there is no 'hold out' data) which shows the same trends we see in the explained variance ratio.

The description of Θ_{inf} is confusing, despite the fact that this parameter seems to be one of the most important to the observed fit. For example, the formal definitions of the models in section 2.1 do not actually include Θ_{inf} anywhere, but at the same the authors include Θ_{inf} into the threshold criterion without describing what this parameter exactly represents. If this parameter is indeed a constant, what is its biological interpretation? Could another potentially ad hoc parameter have been used instead of Θ_{inf} to improve the fit? The reason for post-hoc optimization of this parameter should be in the main text rather than SI. Furthermore the importance of the statement "although Θ_{inf} did change between the fitting and posthoc optimization step, on average the value remains consistent" is unclear because the individual values clearly changed given the improved fits.

We apologize for the confusion here. Θ_{inf} was, in fact, in the description in the equations in section 2.1. It is one component of the threshold of the neuron. Θ_{inf}

is the threshold of a neuron at rest. Thus, θ_{∞} is the threshold uninfluenced by spike or voltage-induced changes. We have added more explanation about why we optimize θ_{∞} in the main text.

We could have chosen to optimize other parameters; however, we chose to optimize this parameter for several reasons. First, it describes the basic excitability of the cell and, thus, could be likely to compensate for imperfections in the model. It is also measured from a much smaller data set (this threshold is measured from a suprathreshold short square: a 3 ms pulse of current) than the rest of the rest of the parameters. R , C , after-spike currents are measured from a linear regression on the training noise set, voltage reset rules are measured from all the spikes in the noise training set, etc. In addition, the short square pulse is given at the start of the entire ephys pipeline protocol which includes many more stimuli than are used in this study. So, it could be that the properties of a neuron change slightly during the recording and thus the measured threshold value is different when the noise stimulus is played. Overall it is likely a very important parameter and one that we had the least confidence in via our methods.

It is likely that if one chose to optimize more parameters that even better results could be found. However, adding more parameters makes the optimization process much longer and prone to overfitting.

The authors actually remove quite a lot of neurons not suitable for fitting to their more complex models (a reduction from >700 for GLIF1 to <300 for some of the more complex models). How useful can these models be if only a minority of the neurons can be meaningfully fit?

The massive reduction of models with GLIF2, 4, and 5 models is not because they cannot be meaningfully fit, it is only because the specific 'triple short square' stimulus protocol that characterizes how the neuron inactivates right after a spike (threshold rises via the spike component of the threshold θ_s) was not presented to the neuron. The electrophysiological pipeline at the Allen Institute has many stimuli provided to the neurons for different experimental programs. They cannot squeeze all the stimuli desired for different programs in one experimental protocol thus we only have the 'triple short square' stimulus for a much smaller number of neurons.

Could the authors comment on why the inhibitory cf. excitatory neurons were fit consistently better?

In our revised manuscript we address this question. We show that it is likely that inhibitory neurons are more stereotypical than excitatory neurons and thus easier to fit. Using a multiple linear regression we show that the spike cut length, and a measure of spike shape reproducibility are statistically significant in predicting spike time performance, whereas, the number of spikes is not.

How much are the more complex models hampered by lack of fit? For instance, GLIF2 performance is actually reduced cf. GLIF1, presumably because the parameter fit is worse. What is the main determinant of the poor fit in this case, Θ_{∞} ? This should be discussed in more detail.

The fact that GLIF2 performs worse is interesting. It shows that adding complexity by including mechanisms fit from the neural voltage waveform does not necessarily increase the spike time performance. This is not because a non-optimal value of θ_{∞} was found: the performance of GLIF2 does still increase after θ_{∞} optimization.

Several aspects could contribute to the reason GLIF2 performs worse. As now stated in the Discussion of our manuscript, "the after-spike currents and the reset rules are fit simultaneously on data which does not isolate their effects. This simultaneous fit could explain why the addition of reset rules to a model without the after-spike currents (GLIF1 \rightarrow GLIF2) hurt the fitting of spike times, while the addition of the same rules along with after-spike currents (GLIF3 \rightarrow GLIF4) improves the fit. Why the addition of after-spike currents in the absence of reset rules (GLIF1 \rightarrow GLIF3) is helpful while the reset rules alone (GLIF1 \rightarrow GLIF2) hurt performance is not clear, but potentially could be caused by the difference in time scales of their effects: the reset rules exert their influence very close to the time of the spike whereas the after-spike currents decay over longer time scales.

Figure 5. It would make more sense to group each type of neuron separately (excitatory, inhibitory, and all neurons). The effect sizes of the models within this groups are more important than the p values and these effect sizes will be easier to ascertain through such a grouping.

In the original manuscript, we did group inhibitory and excitatory cells separately. In the current manuscript we replace Figure 5 with a figure that shows the ability of individual cre lines to reproduce spike times along with inhibitory, excitatory and all neurons. We supply all values in Table 2. This Figure and Table shows how different mechanisms are important for excitatory and inhibitory neurons and how some cre lines are more difficult to fit than others.

The presentation in general is very sloppy. For example, I picked up the following text mistakes during my review, and there could be others: electrophysiological, neurons's, during during, Supplimentary, synthesizing, distinct, and and an, Columns 1 though 5, cells and and neurons, can be rearranges as follows, Can an be seen. (for Brain Science, 2016) is used as the last name of an author, capital and lower theta are used interchangeably in equations, etc. I urge authors to thoroughly clean of the presentation of the manuscript.

We have rewritten the manuscript for clarity and readability.

Reviewer #2 (Remarks to the Author):

The authors fit a neuron model with varying complexity to a large database of electrophysiological recordings and assess whether adding complexity to the model improves its ability to reproduce spike trains in response to noisy input and attempt to cluster the neurons into classes that overlap with the different cell lines based on the parameters of the model. They conclude that added complexity improves performance and model parameters provide a better classification compared to features extracted directly from voltage traces.

While I appreciate the approach to assess the utility of a neuron model with the impressive database of the Allen Institute, I do not see that this study constitutes a substantial addition to the literature.

We hope our revised manuscript has convinced you of our addition to the literature. We have listed our contributions to the literature in our introductory letter. In particular, we implement GLIF modeling methods on a large set of data and find the new scientific result that model parameters can differentiate transgenic lines. This is completely novel. Furthermore, we show that the ability of GLIF parameters to differentiate transgenic lines follows the same trend as their ability to reproduce spike times suggesting that creating simple models that recreate spike times is an effective method to reduce the high dimensional electrophysiological space in order to differentiate cell types without the need for *a-priori* defined features.

My main concern is on methodological issues. The two main results are not supported in a statistical proper way. The improvements of the model performance is assessed using a number of pair comparisons with no apparent correction for alpha error inflation (a Kruskal-Wallis test would be appropriate here). From visual inspection of the boxplots in Figures 5, it is clear that the different models do not notably differ in performance, so even if there are any statistically significant differences, they are of no practical significance.

We have revised our statistics to include an alpha error correction. We start with a Friedman test (as opposed to a Kruskal-Wallis test because our data is pairwise) which shows global differences exist between the different GLIF models' capacity to recreate the spike times of hold-out data. We follow with a Wilcoxon pairwise rank-sum test with a multiple comparison correction using the Benjamini-Hochberg procedure. The p-values were so low to begin with that this correction procedure did not change our ability to reject the null hypothesis: there are statistical differences between the different GLIF levels for inhibitory and excitatory cells. We are not sure how to respond concerning whether these differences are 'notable' or not; they are certainly, statistically significant. We thought it was interesting how much phenomenological mechanisms did or did not improve the ability of models to recreate the spiking behavior of neurons and differentiate transgenic lines.

Similarly, the improvement in clustering the cell lines using the model parameters is claimed to be better compared to clustering based on the voltage trace features, but no quantitative support of this claim is given beyond a superficial examination of Figure 6.

In this version we have included the Adjusted Rand Index to quantify the similarity of different clustering paradigms. In addition, we have included another clustering method (affinity propagation) to substantiate our clustering results.

A second concern is on the significance and novelty of the results. Comparisons of model performance have been performed before (e.g. Jolivet et al. 2008) and the same is true for clustering approaches to neuron classification (e.g. Ardid et al. 2015), so I don't see the exact contribution of this manuscript.

We believe our study will be highly significant to the field. We characterize a large set of neurons under comparable conditions using GLIF models in order to draw broad conclusions concerning their applicability. Jolivet et al 2008 and the INCF have hosted single neuron fitting competitions and compared fitting methods with a limited dataset. Here we focus on applying a set of methods on a large database of responses, producing a resource of models for the community, characterizing the differences in models for different cell types, and exploring how adding phenomenological complexity increases the performance of the traditional LIF model at reproducing spike times and the capacity to classify cell types. We were specifically interested in this model for the reasons listed in our revised manuscript (they are simple, linear in their dynamical equations and thus optimized relatively quickly, the mechanisms added are phenomenologically relevant, and LIF based neurons are widely used in the field and therefore this study will have maximal impact). We think this manuscript will be of specific interest because so much of the field uses traditional LIF models in network models despite the better fit models produced via the neuron fitting competitions and other studies. This study will give intuition concerning just how much improvement in spike time reproduction and the ability to differentiate different cell lines is gained by adding phenomenological complexity.

In addition, we present the completely novel result that GLIF model parameters can differentiate between different transgenic lines without the need for *a priori* defined electrophysiological features.

With respect to the classification of Ardid et al, 2015, we think this is a beautiful paper with a related, but very different focus. They use features of electrophysiological traces of classification, here we use parameters of models: we do not choose features. They justify their clusters by showing their clusters adhere to classifications thought to 'make sense'. Both our study and their study describe the clusters and how we think they relate to classifications that 'make sense'. However, we compare our classification to

molecularly defined cell types. We are the first to show the potential of GLIF parameters to identify transgenic cell lines. We show that with our limited parameters (i.e. we don't use our full set of parameters) we can do as well as subthreshold electrophysiological features, but we do not have to manually select which features are important.

A possible way the authors could go may be a comparison of established neuron models (e.g. in comparison with the GLIF variants) in a statistically sound way (and using more appropriate ways of comparing them, e.g. Bayesian model selection, which incorporates penalties for model complexity). Their rich database of electrophysiological recordings certainly provides a unique resource for such a useful endeavor.

There are many models published in the literature that we could have implemented and tested on our database. However, an evaluation of all models was outside the scope of this study. Instead we wanted to use simple models to achieve new scientific results. We choose our specific models because they are simple, linear in their dynamical equations and thus optimized relatively quickly, the mechanisms added are phenomenologically relevant, and LIF based neurons are widely used in the field and therefore this study will have maximal impact). However, in this version of the manuscript we do report the spike time performance for the morphologically realistic, Hodgkin-Huxley based models created at the Allen Institute available in the Allen Cell Type Database.

It is our understanding that Bayesian model selection is only necessary when there is no 'hold out' data to prove there is not overfitting. Here we test our models on 'hold out' data. Therefore, there is no need to penalize models for having more complexity. However, we did add an additional metric available in the Supplementary Material using the training data and penalize models for complexity using the Akaike Information Criterion (AIC). It follows the same trend as our results on 'hold out' data.

If the increase in spike performance or ability to differentiate cell lines is not compelling to an individual researcher, they can choose to use LIF models with the added comfort of knowing exactly how their performance could compare to other models of increasing phenomenological complexity.

Reviewer #3 (Remarks to the Author):

The models and approach presented in this paper are a valuable resource. The study is conceptually not that different from prior GLIF model development studies, but its use of a very extensive dataset data set sets it apart in implementation. The major result is that GLIF models perform well, and that their parameter space clusters similarly to that of the physiological data.

We are glad this reviewer recognizes the merit of our study. Indeed we draw on the work of many of those before us and show that GLIF models are useful for reproducing spike times and that they can differentiate molecular cell types comparable to subthreshold electrophysiological features.

Major comments.

1. I feel this is more of a resource study than a research article. As a resource, this is in line with the other valuable resources of the Allen Institute. It has brought together a vast amount of data with careful model fitting and testing in a range of GLIF detail.

Indeed, we agree this paper will be a useful resource to the community. However, we do use this resource to discover novel scientific results. Therefore, we would prefer the manuscript to be presented as an article but would be happy with whatever type of publication the Editor believes is most relevant.

I'm not sure there is yet a strong case that having biologically accurate firing patterns in integrate-and-fire models makes a huge difference to network computation. Nevertheless, I expect that the reported models will find wide use. The analysis on how much is gained by different levels of GLIF model will also be useful.

This study can provide one basis for which to start implementing cell types of varying complexity into LIF based network models to assess their computational properties.

2. I don't gain much physiological or computational insight from this study. There is the intriguing point that these models converge during parameter fits, in contrast to the observation that detailed ones have multiple solutions (Prinz and Marder). But I feel that is mostly a reflection of these models rather effectively mapping physiological traces to parameters.

We hope that our revised manuscript gives more physiological and computational insight. As mentioned in our introductory letter, our physiological insights include: 1) that different mechanisms are important for reproducing the spiking behavior of excitatory and inhibitory neurons, and 2) that inhibitory neurons are more stereotypical than excitatory neurons. The computational insights we contribute are that 1) model

parameters can differentiate transgenic lines (This is completely novel), 2) we show that the ability of GLIF parameters to differentiate transgenic lines follows the same trend as their ability to reproduce spike times suggesting that creating simple models that recreate spike times is an effective method to reduce the high dimensional, electrophysiological space in order to differentiate cell types without the need for *a-prori* defined features, and 3) counter to the intuition of many, increasing model complexity by adding phenomenological mechanisms fit directly from the neural voltage waveform, does not necessarily improve the ability of models to reproduce neural spike times.

Minor points:

Figure 1 conveys rather complicated and extensive information, and the figure is overwhelmed by its very long legend. It isn't even referenced in the text. Possibly the authors may consider having a regular introductory section in the main text that refers to the figure? I was looking to see the experimentally observed spiking pattern in panel (b). It is all very well for the GLIFs to converge to a certain spiking pattern, but how close is this to the real thing? This comparison shows up later in Figure 4,9.

We hope Figure 1 in the revised manuscript is more readable. We have added the data to the figure.

Page 3, 4 lines from bottom: typo: "eletrophysiological"

Figure 6 legend: "all optimized model parameters 8"

Figure 6: There is no panel d, though it is referred to in the legend.

Figure 11 legend "Can an be seen"

We have 'cleaned up' the manuscript.

Reviewers' comments:

Reviewer #1 (Remarks to the Author):

In the general the authors have done a great job at addressing my comments on the original version of the manuscript. I have several remaining questions and comments on the revised version of the manuscript and on the authors' main claims.

Specifically, while the fact that 'GLIF parameters are more effective at differentiating cell types associated with transgenic lines than subthreshold electrophysiological features' is indeed a potentially interesting result, more evidence is needed for this result to be convincing.

First, the adjusted rand index is a common measure of partition similarity but is not without problems, and information-theory based metrics (e.g. variation of information) are increasingly used for assessing clustering similarity in a more principled way. See

<https://doi.org/10.1016/j.jmva.2006.11.013> and

<http://www.jmlr.org/papers/volume6/daume05a/daume05a.pdf> for more discussion.

Second, it is unclear how to interpret the displayed values of the adjusted rand index. For instance, while it seems that the clustering based on GLIF1 and GLIF2 is not significant, it is not possible to say this with certainty. It would be useful to have a null comparison, such as a permutation test; e.g. compute partition similarity on clustering of neurons with shuffled labels.

Third, it would be good to illustrate more directly 'the surprising ability of the traditional LIF model to recreate the spike times of a large set of biological neurons', e.g. with a scatter between cluster similarity and explained variance. I couldn't find this plot in the manuscript.

Fourth, the authors discuss the reasons why 'increasing model complexity by adding mechanisms fit from the voltage waveform does not necessarily lead to increased performance in spiking behavior', but again more direct evidence would help. Specifically, it would be useful to have a direct illustration that the models with increased complexity do in fact reproduce subthreshold physiological features with increasing accuracy; e.g. a scatter between variance of subthreshold dynamics, and variance of spike times. If GLIF2 and other models perform better than GLIF1 at describing subthreshold features, then it becomes reasonable to conclude that such features do not directly translate into better prediction of spike times. But if GLIF2 and other models do not perform better than GLIF1 at describing subthreshold features, then the picture becomes more complicated and one has to focus on fitting inaccuracies in model selection.

Relatedly, it would be useful to illustrate this effect more directly using the AIC. At the moment, the authors show the relative AIC of GLIF2, etc. compared to GLIF1. But it would be more straightforward to plot the absolute values of the AIC for both cases; i.e. when considering the goodness of the model for describing spike time variance; and separately when considering the goodness of the model for describing subthreshold membrane variance.

Finally, it is interesting that the spike shape substantially increases the goodness of the clustering of cells into biologically relevant groupings. Can the authors discuss the relevance of this effect, and consider whether future GLIF models could incorporate a spike shape term, in a piecewise linear fashion, to increase the accuracy and completeness of these models even further.

Reviewer #2 (Remarks to the Author):

The authors have significantly improved the manuscript by using the proper statistical methods and quantifying the clustering performance as well as describing their aims and methods more clearly. However, my major concern regarding significance and novelty of the results remain. Overall, I agree with my fellow reviewer that the whole paper read more like a (valuable) resource study than a research article. This impression has not changed in the revised version. I will illustrate this impression using the key results the authors summarized in the rebuttal letter:

A) GLIF clusters can differentiate transgenic lines better than subthreshold electrophysiological features.

-> This is not a surprising result, as the GLIF parameters represent both sub- and suprathreshold aspects of the voltage trace. The fact that only the most complex GLIF variant robustly generates better clustering results may suggest that the number of features used for clustering is the most important factor. A comparison between the different GLIF variants and a matched number of randomly selected features would be fairer. Finally, the Adjusted Rand Index seems small even for the full set of features, casting doubt on the overall usefulness of the clustering.

B) Different phenomenological mechanisms are needed to recreate the firing patterns of excitatory and inhibitory neurons.

-> This may be true, but as one can both see in Figure 5 and 14, the differences are minimal at best. This is what I meant with "notable": Differences that are statistically significant may still be so small that they are of no practical relevance, especially when using a large data set. Effect size measures can be used to quantify the magnitude of the difference - from looking at the figures, I am quite sure they are minimal.

C) The ability of GLIF parameters to differentiate between different transgenic lines follows the same trend as the ability of neurons to reproduce spiking behavior.

-> This could be an indication that better fitted models are better in differentiating transgenic lines. It may also be a byproduct of the increased number of parameters (see above).

D) Quantify the ability of different phenomenological generalizations fit from the data to both recreate spike times and differentiate transgenic lines.

-> See my objections to result A and B

E) Show that inhibitory neurons are easier to fit than excitatory neurons and provided evidence that this is because inhibitory neurons are more stereotypical.

-> Maybe I missed it, but I did not find the mentioned evidence for more stereotypical inhibitory neurons and, more importantly, for the claimed relation to the higher explained variance.

F) Although it may seem intuitive that increasing model complexity by including mechanisms fit to aspects of the voltage waveform would improve spike time reproducibility, this is not always the case.

-> I have not understood under which circumstances there is an improvement, and why.

Reviewer #3 (Remarks to the Author):

Generalized Leaky Integrate-And-Fire Models Classify Multiple Neuron Types Teeter et al.

This revision addresses most of the technical points brought up in the previous version. It does not substantially change the primary concern, that this study is a resource report rather than having a significant research component.

The authors emphasize the finding that "GLIF clusters can differentiate transgenic lines better than subthreshold electrophysiological features." They also provide a list of physiological and computational insights from the model. Together I don't find these to be compelling arguments for a research outcome from this study. I continue to feel that the primary contribution here is as a resource.

Dear Sachin and Reviewers,

We appreciate your input and helping us to improve our manuscript. Sachin requested we streamline the emphasis on the resource aspects of this study. We have done so by adding text to the abstract to describe how these models will contribute to the community. We had already included a similar comment in the last paragraph of the Discussion and the Reviewers have suggested the manuscript already reads as a Resource. We have answered the questions of the Reviewers inline below and have added Figure 7 to the main text and Supplementary Figures 21 through 23 to address these questions. In addition, we show the variability in the binary splitting clustering method via bootstrapping (Supplementary Material, Figure 24). We have highlighted all additions and regions of the manuscript where edits have been made.

Thank you,

Teeter, Et Al.

Reviewers' comments:

Reviewer #1 (Remarks to the Author):

In the general the authors have done a great job at addressing my comments on the original version of the manuscript. I have several remaining questions and comments on the revised version of the manuscript and on the authors' main claims.

Specifically, while the fact that 'GLIF parameters are more effective at differentiating cell types associated with transgenic lines than subthreshold electrophysiological features' is indeed a potentially interesting result, more evidence is needed for this result to be convincing.

First, the adjusted rand index is a common measure of partition similarity but is not without problems, and information-theory based metrics (e.g. variation of information) are increasingly used for assessing clustering similarity in a more principled way. See <https://doi.org/10.1016/j.jmva.2006.11.013> and <http://www.jmlr.org/papers/volume6/daume05a/daume05a.pdf> for more discussion.

We had originally intended to include both the Adjusted Rand Index (ARI) and Variation of Information (VOI). We had rejected the latter because it was clearly sensitive to cluster number in our analysis, consistent with what has been reported before in other studies (e.g. in Vinh et al. 2010). However, thanks to the reviewer's next comment (below), we used the shuffled-label approach to establish a baseline for the VOI, and then used this to calculate the adjusted VOI (as reported in literature), which seems to be independent of the number of clusters. The ARI of shuffled data, by definition (and confirmation – see figure in response to the following comment) is zero, so no further adjustment is needed. We thank the reviewer for emphasizing the importance of including an additional metric, as it strengthens the cluster

comparison results to have both ARI and AVOI in agreement.

Second, it is unclear how to interpret the displayed values of the adjusted rand index. For instance, while it seems that the clustering based on GLIF1 and GLIF2 is not significant, it is not possible to say this with certainty. It would be useful to have a null comparison, such as a permutation test; e.g. compute partition similarity on clustering of neurons with shuffled labels.

By definition, the ARI for the null comparison (permutation test) is zero. We confirmed this by performing a permutation test (please see left panel in the figure below). This suggests that the clustering based on GLIF1 and GLIF2, while resulting in relatively low values for the ARI, is better than expected by chance. We calculated the adjusted VOI, which is the VOI measure adjusted-for-chance and no longer sensitive to the number of clusters (see response to comment above). The adjusted VOI is defined as the shuffled VOI – unshuffled VOI. Note that VOI is a distance measure, and therefore, before the adjustment is implemented, a value of 0 would be perfect similarity and higher numbers would be more similar. Therefore, the shuffled correction is higher (more dissimilar) as can be seen in the right panel below. When the unshuffled VOI is subtracted from the shuffled VOI, the result is a measure where a value of 0 shows no difference from chance partitioning (similar to the ARI), and higher values indicate better agreement among clusters. The use of shuffled labels to calculate the null value of the VOI metric is crucial for its normalization.

Third, it would be good to illustrate more directly 'the surprising ability of the traditional LIF model to recreate the spike times of a large set of biological neurons', e.g. with a scatter between cluster similarity and explained variance. I couldn't find this plot in the manuscript.

The requested scatter plot of cluster similarity (compared to Cre lines) and explained variance of the models' ability to reproduce spike times can now be found in Supplementary

Figure 23 a and b. GLIF1 is better than GLIF2 at reproducing spike times but GLIF2 is better at identifying transgenic lines. There is a correlation between the ability of a model to reproduce spike times and the ability of a model to cluster Cre lines; however, given the small number of data points, it is difficult to claim with statistical certainty. When we perform a linear regression on these data points, we obtain large r values (binary splitting: ARI $r=0.99$, AVOI $r=0.81$, affinity propagation: ARI= 0.90 , AVOI= 0.79). However, the p-values associated with testing whether the slope is significantly different from zero are below standard significance criteria for 3 out of 4 of the clustering similarity metrics (binary splitting: ARI $p=0.14$, AVOI $p=0.17$, affinity propagation: $p=0.01$, AVOI= 0.21). In addition, as pointed out by Reviewer 2 below, perhaps the number of parameters used during clustering could be a contributing factor to 'clusterability' (Supplementary Figure 23 e and f). To avoid drawing strong conclusions, we have softened our statement concerning the trend between spike time performance and clustering performance. We have deleted the statement in our Introduction stating that we observe a trend between spike time performance and clustering performance. We now only point out that the higher level GLIF models do better at clustering and reproducing spike times.

Fourth, the authors discuss the reasons why 'increasing model complexity by adding mechanisms fit from the voltage waveform does not necessarily lead to increased performance in spiking behavior', but again more direct evidence would help. Specifically, it would be useful to have a direct illustration that the models with increased complexity do in fact reproduce subthreshold physiological features with increasing accuracy; e.g. a scatter between variance of subthreshold dynamics, and variance of spike times. If GLIF2 and other models perform better than GLIF1 at describing subthreshold features, then it becomes reasonable to conclude that such features do not directly translate into better prediction of spike times. But if GLIF2 and other models do not perform better than GLIF1 at describing subthreshold features, then the picture becomes more complicated and one has to focus on fitting inaccuracies in model selection.

We have created a new Supplementary Figure 22 to address this question along with including a text description in the Results section. When considering all data from all models, there is a correlation between the ability of a model to reproduce the subthreshold voltage traces of the data and the ability to reproduce its spike times (Figure 22a, black line). However, the median ability of a model level to reproduce spike times does not predict the median values describing the subthreshold voltage match (Figure 22b). Figure 22b shows that models which have spike reset rules fit directly from the data (GLIF2, 4 and 5) better reproduce the subthreshold voltage of the neuron data. Although GLIF3 performs better at reproducing spike times it does less well at reproducing the subthreshold behavior of neurons. In Supplementary Figure 23 c and d, we also show the relationship between the ability of models to reproduce the subthreshold behavior of neurons and the ability of their parameters to differentiate cre-lines

In other words, a model's ability to reproduce subthreshold voltage does not mean it will have increased ability to reproduce spike times.

Relatedly, it would be useful to illustrate this effect more directly using the AIC. At the moment, the authors show the relative AIC of GLIF2, etc. compared to GLIF1. But it would be more straightforward to plot the absolute values of the AIC for both cases; i.e. when considering the goodness of the model for

describing spike time variance; and separately when considering the goodness of the model for describing subthreshold membrane variance.

We respectfully disagree with the Reviewer in regard to the absolute values of AIC. AIC is a tool used for model comparison (K. Burnham, D. Anderson Model Selection and Multimodel Inference Springer, New York (2002)). The interpretation of AIC absolute values is perilous and we feel uncomfortable including them here. Often, AIC is a tool for model selection used when there is not a "hold out" data set available. We explicitly designed the experiments to include a hold-out dataset. We believe that adding absolute AIC values would be more confusing than focusing on the hold-out performance. We have added a plot of the relative subthreshold voltage AIC to Supplementary Figure 18.

Finally, it is interesting that the spike shape substantially increases the goodness of the clustering of cells into biologically relevant groupings. Can the authors discuss the relevance of this effect, and consider whether future GLIF models could incorporate a spike shape term, in a piecewise linear fashion, to increase the accuracy and completeness of these models even further.

The Reviewer brings up an interesting question. Fitting a model to the action potential waveform would require the use of non-linear equations such as in Hodgkin-Huxley type models. Unfortunately, this would destroy the benefits of using linear equations during optimization. However, one could measure the features of the spike waveform and then use them along with GLIF parameters during the clustering (similar to the pairwise fashion you were alluding to). To see if including both the GLIF parameters and the measured spike shape electrophysiological features would improve the identification of Cre lines, we performed both the binary clustering and the affinity propagation on the combined set. The results can be viewed in Figure 7 and Supplementary Figures 20 and we have added text in the Results section to describe these results. Spike shape helps GLIF1 and GLIF2 differentiate Cre lines. The improvement for GLIF3 and GLIF4 is small. This leads us to conclude that subthreshold GLIF4 parameters carry significant information about both the sub and supra threshold electrophysiological features. We include an analysis of clusters with both GLIF 4 and spike feature parameters and this results in the most accurate clusters we report.

Reviewer #2 (Remarks to the Author):

The authors have significantly improved the manuscript by using the proper statistical methods and quantifying the clustering performance as well as describing their aims and methods more clearly. However, my major concern regarding significance and novelty of the results remain. Overall, I agree with my fellow reviewer that the whole paper read more like a (valuable) resource study than a research article. This impression has not changed in the revised version. I will illustrate this impression using the key results the authors summarized in the rebuttal letter:

*A) GLIF clusters can differentiate transgenic lines better than subthreshold electrophysiological features.
-> This is not a surprising result, as the GLIF parameters represent both sub- and suprathreshold aspects*

of the voltage trace. The fact that only the most complex GLIF variant robustly generates better clustering results may suggest that the number of features used for clustering is the most important factor. A comparison between the different GLIF variants and a matched number of randomly selected features would be fairer. Finally, the Adjusted Rand Index seems small even for the full set of features, casting doubt on the overall usefulness of the clustering.

The reviewer raises two excellent points here, and we have included additional information in the manuscript to address this. First, we have included the mean cluster comparison metrics for random subsets of features, matched to the number of parameters for each of the GLIF models. This is shown as lighter colored lines in the clustering figure; whereas there is an increase in the metrics from 5 to 7 parameters (GLIF1 has 5 parameters, GLIF2 and GLIF3 have 7 each), there is no substantial increase from 7 to 9 (GLIF4 has 9 parameters). This suggests that, although the number of parameters may play some role in the improvement of clustering with GLIF parameters, it is not wholly responsible for the better clustering obtained by the successive GLIF models.

Second, to address the relatively low values of the Adjusted Rand Index (and a second measure introduced in this revision, the Adjusted Variation of Information), we calculated the ARI and AVOI for transcriptomically-derived types (from Tasic et al. 2016), using the same subset of Cre lines. The resulting values for these clusterings are 0.30 (for the ARI) and 2.96 (for the AVOI), which are better than those obtained by the GLIF clusterings, but not overwhelmingly so. The main explanation for this is that the transgenic lines themselves label overlapping subsets of cells, and are thus not a perfect representation of distinct cell types. However, in the absence of direct transcriptome-ephys data obtained from the same sets of cells, the Cre lines are the best approximation of the “ground truth”, imperfect as they are. This is mentioned in the text, and the ARI and AVOI values for the transcriptomically-derived types are included in order to put the values of the ARI and AVOI for the GLIF clustering in context.

B) Different phenomenological mechanisms are needed to recreate the firing patterns of excitatory and inhibitory neurons.

-> This may be true, but as one can both see in Figure 5 and 14, the differences are minimal at best. This is what I meant with "notable": Differences that are statistically significant may still be so small that they are of no practical relevance, especially when using a large data set. Effect size measures can be used to quantify the magnitude of the difference - from looking at the figures, I am quite sure they are minimal.

While the effects are minimal, they are statistically significant. We believe it is biologically realistic to expect some differences. Different neurons certainly have different ion channels and passive characteristics that allow them to have different spiking behavior. It makes sense that we would see these differences between inhibitory and excitatory and some cre-lines in the phenomenological mechanisms.

C) The ability of GLIF parameters to differentiate between different transgenic lines follows the same trend as the ability of neurons to reproduce spiking behavior.

-> This could be an indication that better fitted models are better in differentiating transgenic lines. It may also be a byproduct of the increased number of parameters (see above).

The Reviewer brings up an interesting point concerning the relationship between the number of parameters used in clustering and clustering ability. In Supplementary Figure 23 e and f, we supply the relationship between number of parameters fit and the ability to differentiate Cre lines. Overall, there is a relationship between number of parameters and the ability of the algorithms to differentiate Cre lines. Although the same number of parameters are used in the clustering, GLIF3 parameters appear to be more useful (when spike features are not used) at differentiating Cre lines than GLIF2 parameters in binary splitting and affinity propagation.

We have replaced our statement, "The ability of GLIF parameters to differentiate between different transgenic lines follows the same trend as the ability of neurons to reproduce spiking behavior", by, "5) Parameters obtained from fitting neurons with GLIF models are useful in classifying cell types: higher level GLIF parameters are more effective at differentiating cell types associated with transgenic lines than subthreshold electrophysiological features."

D) Quantify the ability of different phenomenological generalizations fit from the data to both recreate spike times and differentiate transgenic lines.

-> See my objections to result A and B

See our response to A and B above.

E) Show that inhibitory neurons are easier to fit than excitatory neurons and provided evidence that this is because inhibitory neurons are more stereotypical.

-> Maybe I missed it, but I did not find the mentioned evidence for more stereotypical inhibitory neurons and, more importantly, for the claimed relation to the higher explained variance.

This is discussed in the second to the last paragraph in section 2.3 Model Performance (on page 6). As mentioned in the text, the data are shown Supplementary Material, Figure 10. Briefly, we show that spike cut length and spike reproducibility (as measured by the standard error when measuring spike cut length) are more indicative of a model's ability to reproduce spike times than the number of spikes fired by a neuron.

F) Although it may seem intuitive that increasing model complexity by including mechanisms fit to aspects of the voltage waveform would improve spike time reproducibility, this is not always the case.

-> I have not understood under which circumstances there is an improvement, and why.

In this revision, we included a paragraph (paragraph 3 of the 2.3 Model Performance section) in the main text and Figure 22 in the Supplementary Material to address the ability of the models to reproduce the subthreshold voltage. These results reiterate that the ability of the model to reproduce subthreshold voltage does not necessarily translate into the ability of a model to reproduce spike times. We added additional text (paragraph 4 of the Discussion section) to more explicitly describe possible reasons that mechanisms fit directly from the data do not necessarily lead to increased spike performance.

Reviewer #3 (Remarks to the Author):

*Generalized Leaky Integrate-And-Fire Models Classify Multiple Neuron Types
Teeter et al.*

This revision addresses most of the technical points brought up in the previous version. It does not substantially change the primary concern, that this study is a resource report rather than having a significant research component.

The authors emphasize the finding that "GLIF clusters can differentiate transgenic lines better than subthreshold electrophysiological features." They also provide a list of physiological and computational insights from the model. Together I don't find these to be compelling arguments for a research outcome from this study. I continue to feel that the primary contribution here is as a resource.

We thank the Reviewer for taking the time to consider our manuscript. We are happy they believe we have made a valuable resource for the community.

REVIEWERS' COMMENTS:

Reviewer #1 (Remarks to the Author):

I thank the authors for making additional revisions to the manuscript. In general, most of my comments have been satisfactorily addressed and from my perspective the manuscript is essentially acceptable for publication.

Despite this, there is still one quirk which makes interpretation of the models somewhat confusing. In my previous comments I noted that,

“It would be useful to have a direct illustration that the models with increased complexity do in fact reproduce subthreshold physiological features with increasing accuracy. If GLIF2 and other models do not perform better than GLIF1 at describing subthreshold features, then the picture becomes more complicated and one has to focus on fitting inaccuracies in model selection.”

The authors essentially replied that “a model's ability to reproduce subthreshold voltage does not mean it will have increased ability to reproduce spike times” which is fine but does not address one of the points I was trying to make. Essentially it seems here that the authors do not show a monotonic increase in the goodness of fit (for some metric – either for subthreshold voltage or for spike times) that one would expect with a monotonic increase in model complexity. For example, Figure 22b shows that GLIF1 performs intermediately to GLIF2 and GLIF3; seemingly a counterintuitive result given that it is the simplest model and so should perform the worst. I believe a discussion of this effect is warranted in the final version of the paper.

REVIEWERS' COMMENTS:

Reviewer #1 (Remarks to the Author):

I thank the authors for making additional revisions to the manuscript. In general, most of my comments have been satisfactorily addressed and from my perspective the manuscript is essentially acceptable for publication.

Despite this, there is still one quirk which makes interpretation of the models somewhat confusing. In my previous comments I noted that,

“It would be useful to have a direct illustration that the models with increased complexity do in fact reproduce subthreshold physiological features with increasing accuracy. If GLIF2 and other models do not perform better than GLIF1 at describing subthreshold features, then the picture becomes more complicated and one has to focus on fitting inaccuracies in model selection.”

The authors essentially replied that “a model's ability to reproduce subthreshold voltage does not mean it will have increased ability to reproduce spike times” which is fine but does not address one of the points I was trying to make. Essentially it seems here that the authors do not show a monotonic increase in the goodness of fit (for some metric – either for subthreshold voltage or for spike times) that one would expect with a monotonic increase in model complexity. For example, Figure 22b shows that GLIF1 performs intermediately to GLIF2 and GLIF3; seemingly a counterintuitive result given that it is the simplest model and so should perform the worst. I believe a discussion of this effect is warranted in the final version of the paper.

We thank the Reviewer for drawing attention to the difficult concepts in our manuscript. We agree, it is interesting that there is not a monotonic increase in subthreshold voltage or in spike time reproduction with an increase in model complexity. In the last version of the manuscript we did discuss model complexity and give several potential reasons model complexity does not necessarily lead to higher performance in either subthreshold voltage or spike time reproduction. In the updated version of the manuscript, we have underlined the previous text referring to this topic. In addition, we have added additional highlighted text meant to help guide the reader.